# Crystal structures of fukutin-related protein (FKRP), a ribitol-phosphate transferase related to muscular dystrophy

Naoyuki Kuwabara [1,8], Rieko Imae[2,8], Hiroshi Manya[2], Tomohiro Tanaka[3], Mamoru Mizuno[3], Hiroki Tsumoto[4], Motoi Kanagawa[5], Kazuhiro Kobayashi[5], Tatsushi Toda[5,6], Toshiya Senda[1,7], Tamao Endo[2]* & Ryuichi Kato [1,7]*

α-Dystroglycan (α-DG) is a highly-glycosylated surface membrane protein. Defects in the O-mannosyl glycan of α-DG cause dystroglycanopathy, a group of congenital muscular dystrophies. The core M3 O-mannosyl glycan contains tandem ribitol-phosphate (RboP), a characteristic feature first found in mammals. Fukutin and fukutin-related protein (FKRP), whose mutated genes underlie dystroglycanopathy, sequentially transfer RboP from cytidine diphosphate-ribitol (CDP-Rbo) to form a tandem RboP unit in the core M3 glycan. Here, we report a series of crystal structures of FKRP with and without donor (CDP-Rbo) and/or acceptor [RboP-(phospho-)core M3 peptide] substrates. FKRP has N-terminal stem and C-terminal catalytic domains, and forms a tetramer both in crystal and in solution. In the acceptor complex, the phosphate group of RboP is recognized by the catalytic domain of one subunit, and a phosphate group on O-mannose is recognized by the stem domain of another subunit. Structure-based functional studies confirmed that the dimeric structure is essential for FKRP enzymatic activity.

[1] Structural Biology Research Center, Institute of Materials Structure Science, High Energy Accelerator Research Organization, Tsukuba, Ibaraki 305-0801, Japan. [2] Molecular Glycobiology, Research Team for Mechanism of Aging, Tokyo Metropolitan Geriatric Hospital and Institute of Gerontology, Itabashi-ku, Tokyo 173-0015, Japan. [3] Laboratory of Glyco-organic Chemistry, The Noguchi Institute, Itabashi-ku, Tokyo 173-0003, Japan. [4] Proteome Research, Research Team for Mechanism of Aging, Tokyo Metropolitan Geriatric Hospital and Institute of Gerontology, Itabashi-ku, Tokyo 173-0015, Japan. [5] Division of Molecular Brain Science, Kobe University Graduate School of Medicine, Kobe, Hyogo 650-0017, Japan. [6] Department of Neurology, Graduate School of Medicine, The University of Tokyo, 7-3-1 Hongo, Bunkyo-ku, Tokyo 113-8655, Japan. [7] School of High Energy Accelerator Science, SOKENDAI, Tsukuba, Ibaraki 305-0801, Japan. [8] These authors contributed equally: Naoyuki Kuwabara, Rieko Imae. *email: endo@tmig.or.jp; ryuichi.kato@kek.jp

The post-translational glycosylation of proteins by *O*-linked mannose (Man) is a conserved modification from yeast to humans and has been shown to be necessary for proper development and growth. In recent years, *O*-mannosyl glycans have been demonstrated to play critical roles in cellular interaction-based pathologies, including congenital muscular dystrophies and cancers[1–3].

α-Dystroglycan (α-DG) is a central component of the dystrophin-glycoprotein complex (DGC) and serves as a receptor protein for laminin in the basement membrane[4]. α-DG is known to be heavily decorated with *O*-linked glycans, including mucin-type *O*-linked *N*-acetylgalactosamine (GalNAc) and *O*-mannosyl glycans. Currently, the structures of *O*-mannosyl glycans are classified into three types based on the linkage of *N*-acetylglucosamine (GlcNAc) to the mannose (Man) residue: core M1 (GlcNAcβ1-2Man), core M2 [GlcNAcβ1-2(GlcNAcβ1-6)Man], and core M3 (GalNAcβ1-3GlcNAcβ1-4Man)[5]. The biosynthesis of *O*-mannosyl glycans is initiated by protein *O*-mannosyl-transferases (POMT1 and POMT2) at Ser/Thr residues in the ER[6]. After the *O*-mannosylation, syntheses of cores M1 and M2 proceed in the Golgi. Protein *O*-linked mannose β-1,2-*N*-acetylglucosaminyltransferase 1 (POMGNT1) forms core M1 using UDP-GlcNAc as a donor substrate[7]. The core M2 structure is formed sequentially through actions of POMGNT1 and β-1,6-*N*-acetylglucosaminyltransferase IX(VB) [GNT-IX(VB)], an enzyme that catalyzes the formation of the GlcNAcβ1-6Man linkage[8,9].

On the other hand, synthesis of core M3 requires several steps of reactions in the ER after the *O*-mannosylation. Protein *O*-linked mannose β-1,4-*N*-acetylglucosaminyltransferase 2 (POMGNT2, formerly GTDC2 or AGO61), β-1,3-*N*-acetylgalactosaminyltransferase 2 (B3GALNT2), and protein-*O*-mannose kinase (POMK) sequentially modify the *O*-Man residue, forming a phospho-core M3 structure [GalNAcβ1-3GlcNAcβ1-4(phospho-6)Man][5]. The dystroglycan modified with the synthesized phospho-core M3 is transported to the Golgi apparatus. Then, fukutin (FKTN) and fukutin-related protein (FKRP) add a tandem ribitol-phosphate (RboP), and ribitol xylosyltransferase 1 (RXYLT1, formerly TMEM5) and β-1,4-glucuronyltransferase 1 (B4GAT1) add a single GlcAβ1-4Xylβ1-4 unit to the phospho-core M3[10–13]. Finally, a GlcA-Xyl (GlcAβ1-3Xylα1-3) repeat is added by α-1,3-xylosyl- and β-1,3-glucuronyltransferase 1 (LARGE1) to complete the synthesis of the core M3 glycan on α-DG[14]. The multivalency of the GlcA-Xyl repeat of core M3 glycan of α-DG has been thought to regulate the affinity with laminin α2 on the sarcolemma membrane[15,16]. Defects of core M3 glycan synthesis therefore lead to a loss of normal interactions between α-DG and laminin α2, resulting in several diseases such as muscular dystrophy. Indeed, mutations of FKRP, which is involved in synthesis of a tandem RboP structure, cause congenital muscular dystrophy;[17–23] more than 15 missense mutations in the *FKRP* gene were found in LGMD2I (limb-girdle muscular dystrophy type 2I), MDDGB5 [muscular dystrophy-dystroglycanopathy (congenital with or without mental retardation, type B, 5)], and severe muscle-eye-brain disease/Walker-Warburg syndrome (MEB/WWS) patients (OMIM ID of FKRP; 606596). These facts suggest that the tandem RboP structure is critical to synthesize the functional core M3 glycan.

Despite its functional importance, RboP was not found in mammals until recently. RboP is well known as a major building block of the cell wall in Gram-positive bacteria[24]. In bacteria, RboP transferases transfer a RboP moiety from cytidine diphosphate-ribitol (CDP-Rbo) to form a RboP polymer in the biosynthesis of teichoic acid[24]. However, no RboP polymers have been found in mammals; only tandem RboP was identified in the core M3 glycan structure in 2016[11]. Our earlier studies have revealed that FKTN and FKRP are involved in the synthesis

of the tandem RboP. FKTN transfers the first RboP to the third-position of GalNAc from CDP-Rbo, which is synthesized from RboP and CTP by isoprenoid synthase domain-containing (ISPD)[11,25], and FKRP transfers the second RboP to the 1st-position of the first RboP[11]. The synthesis of a tandem RboP unit seems to be highly regulated by the strict substrate specificities of FKTN and FKRP. Since the appropriate synthesis of the tandem RboP is required for the normal function of core M3 glycan, the substrate recognition and catalytic mechanisms of FKTN and FKRP have attracted the attention of many researchers. Furthermore, while FKRP forms a dimer (or oligomer) in vivo[26], its functional implication remains elusive.

In this study, to analyze the ligand recognition mechanism of FKRP, we determined the crystal structures of FKRP with substrates including CDP-Rbo, CMP, and RboP-(phospho-)core M3 peptide. Our structural and biochemical analyses revealed the acceptor substrate recognition mechanism by dimer FKRP: the phosphate group of RboP is recognized by the catalytic domain of one subunit, and a phosphate group on *O*-mannose is recognized by the stem domain of another subunit.

## Results

**Overall structure**. Initially, we determined the crystal structure of human FKRP without substrates. A soluble form of FKRP (sFKRP: residues 45 to C-terminus of FKRP, 451 amino acid residues in total) was purified and crystallized, and the X-ray structures of the $Mg^{2+}$ and $Ba^{2+}$-bound forms were determined (see Methods and Supplementary Table 1). The asymmetrical unit contains four sFKRP subunits, which are designated as subunits (or chains) A, B, C, and D (Fig. 1a). Buried surface areas calculated by ePISA[27] (Supplementary Table 2) suggested that the four sFKRP subunits form a homo-tetramer. While homo-tetrameric protein complexes usually have a point group symmetry of *222*, the sFKRP tetramer has a point group symmetry of *2* ($C_2$). The tetramer is composed of two identical protomeric dimers, dimers AB and CD, and subunits A and B (C and D) are related by a local two-fold axis in the protomeric dimer (Fig. 1b). The two protomeric dimers are related by a two-fold axis in the tetramer with a buried surface area of 1,642 Å[2]. For the dimer-dimer interface, the contribution of the two stem domains is the largest (730 Å[2]), followed by the contribution of the stem and catalytic domains (504 Å[2]), and finally that of the two catalytic domains (336 Å[2]). The tetramer can therefore be considered as a dimer of protomeric dimers.

To examine the oligomeric state of sFKRP in solution, the solution structure of the sFKRP was analyzed by small-angle X-ray scattering with size exclusion chromatography (SEC-SAXS) (Fig. 2a, b). The SEC analysis showed a single peak, suggesting that sFKRP is in a monodisperse state in solution. The SAXS analysis revealed that the radius of gyration from the Guinier plot was 44 Å (average value from 7.48 to 8.17 ml of retention volume). We then performed the *P*(r) function analysis at around the peak fraction. The molecular weight was calculated as 214 kDa from the Porod volume of each *P*(r) function, suggesting that the sFKRP forms a tetramer in solution.

**Domain structures**. As previously predicted, sFKRP is composed of two domains, the stem (residues 45–287) and catalytic (residues 288–495) domains[26]. The overall structure of the stem domain is similar to that of the catalytic domain of polypeptide GalNAc transferase 10 (ppGalNAcT-10) (residues Asn146 to Val453)[28]. However, the HxD motif of ppGalNAcT-10, which is critical to its catalytic reaction, was not conserved. Therefore, the stem domain of FKRP would not have glycosyltransferase activity despite its overall structural similarity to ppGalNAcT-10.

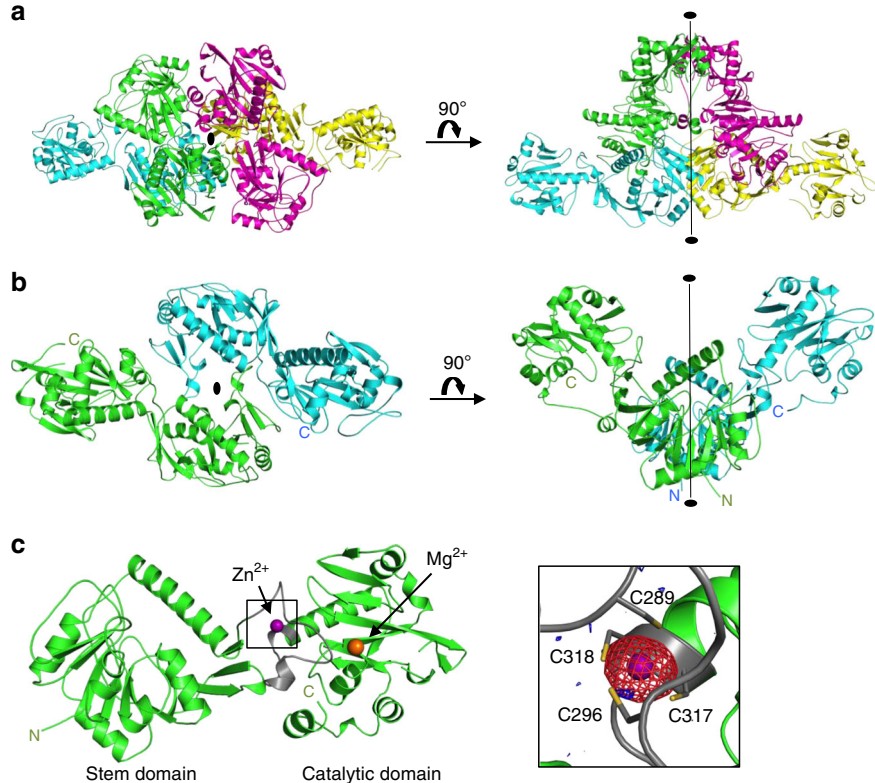

**Fig. 1 Crystal structure of the sFKRP.** All models were prepared using an Mg$^{2+}$ bound structure. **a** Crystal structure of sFKRP showing four subunits in the asymmetrical unit. The subunits are colored green, blue, red, and yellow, respectively. The two-fold axis of the tetramer is shown as a black ellipse and a line. **b** The protomeric dimer of sFKRP. The local two-fold axis of the protomeric dimer is shown as a black ellipse and a line. **c** Monomer structure of sFKRP. Zn$^{2+}$ and Mg$^{2+}$ are shown in purple and orange, respectively. The zinc finger loop (G288 to C318) is shown in gray. The anomalous difference Fourier maps around the zinc finger for the peak data set (red mesh) and the low remote data set (blue mesh) at a resolution of 2.41 Å are shown in the inset. The contour levels of the peak and the low remote are 5.0 and 3.5 σ, respectively. Labels N and C indicate the N- and C-terminus of sFKRP, respectively.

The catalytic domain adopts the nucleotidyltransferase fold (NTase fold), which is centered on a twisted β-sheet surrounded by seven α helices (Fig. 1c and Supplementary Fig. 1a). The core region of the NTase fold superfamily is composed of four α helices and three β sheets (α-β-α-β-α-β-α) and possesses three conserved acidic amino acid residues (Asp362, Asp364, Asp416) and an hG[GS] motif (h represents a hydrophobic amino acid residue; Gly344, Gly345, Ser346)[29]. The three conserved aspartic acids have been considered to be catalytic residues[29]. The core region of the catalytic domain is formed by residues Ala321 to Ala473 and similar to other NTase family proteins, such as Antibiotic-Resistance Factor ANT(2")-Ia (rmsd 2.78 Å with 117 Cα atoms) and lincosamide antibiotic adenylyltransferase LnuA (rmsd 2.48 Å with 110 Cα atoms) (Supplementary Fig. 1a). The three acidic residues and hG[GS] motif are highly conserved in FKRP orthologues (Supplementary Fig. 2). Although FKRP exhibits RboP transferase, not NTase activity, our crystal structure clearly shows that FKRP belongs to the NTase superfamily as proposed previously[29].

In the catalytic domain, a Zn$^{2+}$ that coordinates Cys289, Cys296, Cys317, and Cys318 with distances of 2.3–2.4 Å was found (Fig. 1c). The Zn$^{2+}$ binds together an α7 helix and a loop region (Gly288 to Cys318), and stabilizes the loop. We designated this loop as the zinc finger loop (Fig. 1c, gray). The four Cys residues are highly conserved in FKRP orthologues (Supplementary Fig. 2). The functional importance of the C4-type zinc finger is revealed by the fact that the Cys318Tyr mutation caused FKRP-related WWS, which is the most severe form of dystroglycanopathy[30].

**Relation between disease-related mutations and FKRP oligomerization.** As described in the Introduction, many disease-related mutations have so far been reported and the mutation sites are spread over the entire sequence of FKRP[31–33]. Some of these mutation sites are located at domain-domain and others at subunit-subunit interfaces (Supplementary Fig. 3). We considered that these mutations may change the oligomeric states of FKRP. To investigate this hypothesis, Tyr88Phe, Ser221Arg, and Leu276Ile were selected from the OMIM database. Tyr88 is located at the subunit-subunit interface of the protomeric dimer. Since the hydroxyl group of Tyr88 forms a hydrogen bond with the main chain carbonyl oxygen of Thr304 in an adjacent subunit, Tyr88Phe mutation may change the inter-subunit interaction. Since Ser221 is also located near the inter-subunit interface of the protomeric dimer, replacement of Ser221 with Arg would alter the inter-subunit interaction. Leu276 is in the stem domain and interacts with the catalytic domain in the same subunit. Replacement of Leu276 with Ile would affect the arrangement of the two domains. To examine the effects of these mutations, their oligomerization states and glycosyltransferase activity were analyzed. In the SEC analysis, the apparent molecular weight of the mutated proteins was lower than that of the WT, suggesting dissociation of a subunit(s) from the tetramer (Fig. 2c). The glycosyltransferase activities of the Tyr88Phe, Ser221Arg, and Leu276Ile mutants were approximately 20%, 5%, and 50% of the wild-type enzyme, respectively (Fig. 2d, e). The activities of these mutants seem to correlate with the phenotype severity observed in patients[31,33,34]. In the case of the Leu276Ile mutant, higher molecular weight aggregates may help to partially preserve the

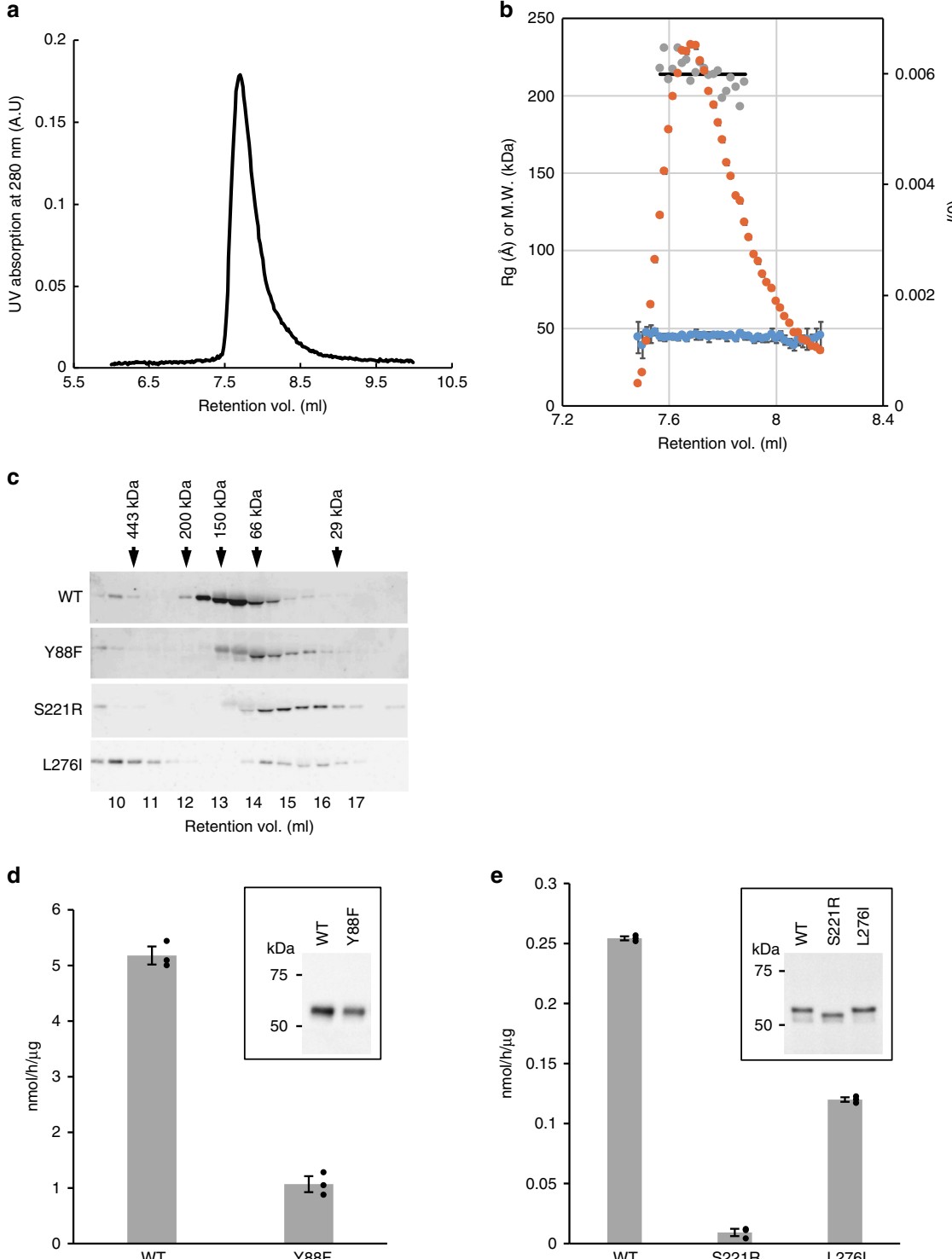

**Fig. 2 Oligomerization analysis.** Oligomerization states in solution and enzymatic activities of wild-type and disease-related mutants of sFKRP were studied. SEC-SAXS profiles of wild-type sFKRP are shown in (**a**) (UV absorption) and (**b**). In (**b**), the SAXS intensity $I(0)$ (orange), radii of gyration ($Rg$) (blue), and estimated molecular weight from Porod volume (gray) are plotted. **a** Black line indicates the average of estimated molecular weight from the Porod volumes. **c** Expressed sFKRP proteins were analyzed by usual SEC and detected by SDS-PAGE (5–20% acrylamide; ATTO) followed by Western blotting. Arrows indicate fractions at which protein markers were eluted. **d**, **e** Enzymatic activities of sFKRP with CDP-Rbo and the RboP-(phospho-)core M3 peptide. Insets: immunoblot analyses of sFKRP proteins to normalize input sFKRP. **d** sFKRP (WT and Y88F) immunoprecipitated from the culture supernatant were used as the enzyme sources. **e** Cell lysates expressing sFKRP (WT, S221R, and L276I) were used as the enzyme sources. Average values ± SE of three independent experiments are shown. Each dot represents one data point. Source data are provided as a Source Data file.

enzymatic activity. Taken together, these results showed that the oligomerization state of FKRP is likely to affect the glycosyl-transferase activity.

**A metal ion-binding site in substrate-free form**. NTase super-family proteins require divalent metal ions for their catalytic reactions. ANT(2")-Ia complexed with AMPCPP and gentamicin C1 has two $Mn^{2+}$ in the active site[35]. A similar metal-binding site was observed in the sFKRP structure. In the substrate-free form of sFKRP, two $Mg^{2+}$ were found in the active site of the catalytic domain (Fig. 3a, and Supplementary Table 3). In the first $Mg^{2+}$ binding site (binding site I), Asp360, Asp362 and Asp364 weakly interact with the $Mg^{2+}$. This $Mg^{2+}$ could not be modeled in subunits C and D due to the weak interaction. On the other hand, the second $Mg^{2+}$ binding site (binding site II) is more stable. The $Mg^{2+}$ at site II has an octahedral coordination sphere and coordinates Asp362 and five water molecules with coordination distances of 2.0–2.3 Å. In addition, Asp364 and Asp416 form hydrogen bonds with water molecules that coordinate the $Mg^{2+}$ and stabilize the coordination sphere. Due to this tight interaction with the protein, the $Mg^{2+}$ is observed in every subunit. It is of note that the crystal structure of the $Ba^{2+}$ bound form showed electron densities for $Ba^{2+}$ only at site II of subunit B (Supplementary Table 3). Since site I has long coordination distances (Supplementary Table 3), it seems to be capable of accommodating $Ba^{2+}$ with ionic radius of approximately 1.4 Å, which is larger than the radii size of 0.7 Å of $Mg^{2+}$ (Fig. 3b, Supplementary Figs. 7 and 8b). However, site II has significantly shorter coordination distances than site I, and thus it seems to be impossible to bind $Ba^{2+}$ without destroying the coordination sphere. These differences in the metal-binding site seem to affect the enzymatic activity.

We then examined the divalent cation requirement for the enzymatic activity of sFKRP. The enzyme assay revealed that sFKRP activity was stimulated by $Mg^{2+}$ or $Mn^{2+}$, but not by $Ba^{2+}$ or $Ca^{2+}$ (Supplementary Fig. 4). While sFKRP is active with $Mg^{2+}$ or $Mn^{2+}$ in vitro, we did not have experimental data for the metal ions of FKRP in cells. Since we could not obtain crystals of the $Mn^{2+}$-bound form, we used the $Mg^{2+}$-bound form for structural and biochemical analysis. Mutational analysis suggested that the three Asp residues that stabilize the $Mg^{2+}$ binding are critical to the enzymatic activity (Fig. 3f). In addition, Asp360, which coordinates the $Mg^{2+}$ at binding site I, is indispensable to the enzymatic activity (Fig. 3f).

**CDP-Rbo (donor) recognition**. To understand the molecular basis of the unique substrate recognition and catalytic reaction of FKRP, we determined the crystal structures of the sFKRP-CDP-Rbo complex with $Ba^{2+}$ and the sFKRP-CMP complex with $Mg^{2+}$ at 2.23 Å and 2.60 Å resolution, respectively (Fig. 3c, d, and Supplementary Table 1). We tried to obtain the complex structure of sFKRP-CDP-Rbo with $Mg^{2+}$ but could not. In the presence of $Mg^{2+}$, electron densities could only be observed for the CMP moiety of CDP-Rbo; those for the RboP moiety were missing (Fig. 3d). Since FKRP is active with the $Mg^{2+}$, CDP-Rbo would be catalyzed and only the CMP moiety would remain under the $Mg^{2+}$-containing condition. Since the enzymatic activity does not appear in the presence of $Ba^{2+}$ (Supplementary Fig. 4), the whole substrate remained under the $Ba^{2+}$-containing condition. In the CDP-Rbo complex structure, the α phosphate residue of CDP-Rbo seems to interact with the $Ba^{2+}$ ion in site I (Fig. 3c, and Supplementary Table 3). The loop region from Leu433 to Gln437, which is visible only in the CDP-Rbo complex structure, is involved in the interaction with the Rbo moiety (Fig. 3c, left gray loop). The CDP-Rbo complex structure suggests that the metal

ion at site II, which could not be observed in the crystal structure, also interacts with the donor molecule CDP-Rbo (Supplementary Fig. 8a). In canonical NTase, however, the metal ions at sites I and II interact with the donor and the acceptor molecules, respectively (Supplementary Fig. 1b, right panel).

**CDP-Rbo binding evokes conformational changes**. Upon CDP-Rbo binding, significant conformational changes of FKRP were induced in three regions: the zinc finger loop (Gly288 to Cys318), the Rbo-interacting loop (Leu433 to Gln437), and a C-terminal fragment (Pro481 to C-terminus). The most significant structural change occurred in the C-terminal fragment of subunits A and C (Fig. 3e). In the substrate-free form, residues from Pro481 to Thr492 of subunits A and C are assumed to adopt an α helical conformation (Fig. 3a) and interact with subunit B and D, respectively. Upon CDP-Rbo binding, the α helix is unfolded and Gln482 makes an interaction with the CDP moiety of the CDP-Rbo (Fig. 3e). Moreover, a small hydrophobic patch was formed in the C-terminal fragment by Tyr483, Pro484, and Leu488 interacting with Asp303, Leu308, Tyr309, and Trp313 in the zinc finger loop by hydrophobic interaction and a hydrogen bond (Fig. 3e). These interactions shift the zinc finger loop toward the CDP moiety of CDP-Rbo; the shift was especially large around the residues Gly298 to Pro305. Then, Asp303 in the zinc finger loop forms a hydrogen bond with His435 in the Rbo-interacting loop, which is disordered in the substrate-free form. The hydrogen bond between Gln437 and CDP-Rbo seems to further stabilize the Rbo-interacting loop, covering the bound CDP-Rbo (Supplementary Fig. 10).

**RboP-(phospho-)core M3 (acceptor) recognition**. FKRP recognizes RboP-(phospho-)core M3 as the RboP acceptor and transfers a RboP from CDP-Rbo to form the tandem RboP structure[11]. To reveal the acceptor recognition mechanism, the crystal structure of sFKRP in complex with an acceptor glyco-peptide was determined. Crystals of the $Ba^{2+}$-bound form were soaked in a solution containing synthetic RboP-(phospho-)core M3 peptide [AT(RboP-3GalNAcβ1-3GlcNAcβ1-4(phospho-6)Manα1-)PAPVAAIGPK] (hereafter acceptor glycopeptide) and the crystal structure was determined at 2.47 Å resolution. While the electron density for a part of the acceptor glycopeptide was observed in the crystal structure (Fig. 4a–c), the Rbo moiety and the peptide part could not be observed, probably due to their mobile characters. Interestingly, the phospho-3GalNAcβ1-3GlcNAcβ1-4(phospho-6)Man structure in the acceptor glycopeptide is recognized by the two subunits in the protomeric dimer (subunits C and D). Lys256 and His252 in the stem domain of subunit D interacts with the phosphate group of phospho-6Man (Fig. 4c). The phosphate group of the other side of the acceptor glycopeptide, which is the phosphate group of RboP, interacts with Arg295 and Val300 in the catalytic domain of subunit C. The N-acetyl group of the GalNAc moiety interacts with Thr299 and His412 in subunit C. It is of note that the phosphate group of RboP in the acceptor glycopeptide is located near the metal-binding site II of the active site. Since the distance between the β phosphate group of CDP-Rbo and the phosphate group of the RboP moiety was 9.7 Å, it would seem to be possible to accommodate the missing Rbo moiety of the acceptor glycopeptide between the two phosphate groups. These results suggest that both the catalytic and stem domains from two distinct subunits are required for the acceptor glycopeptide recognition. However, we could not elucidate the detailed interaction between them; since the affinity for the acceptor glycopeptide was relatively low, the occupancy of the soaked acceptor glycopeptide in the crystal was not sufficient for concrete analysis.

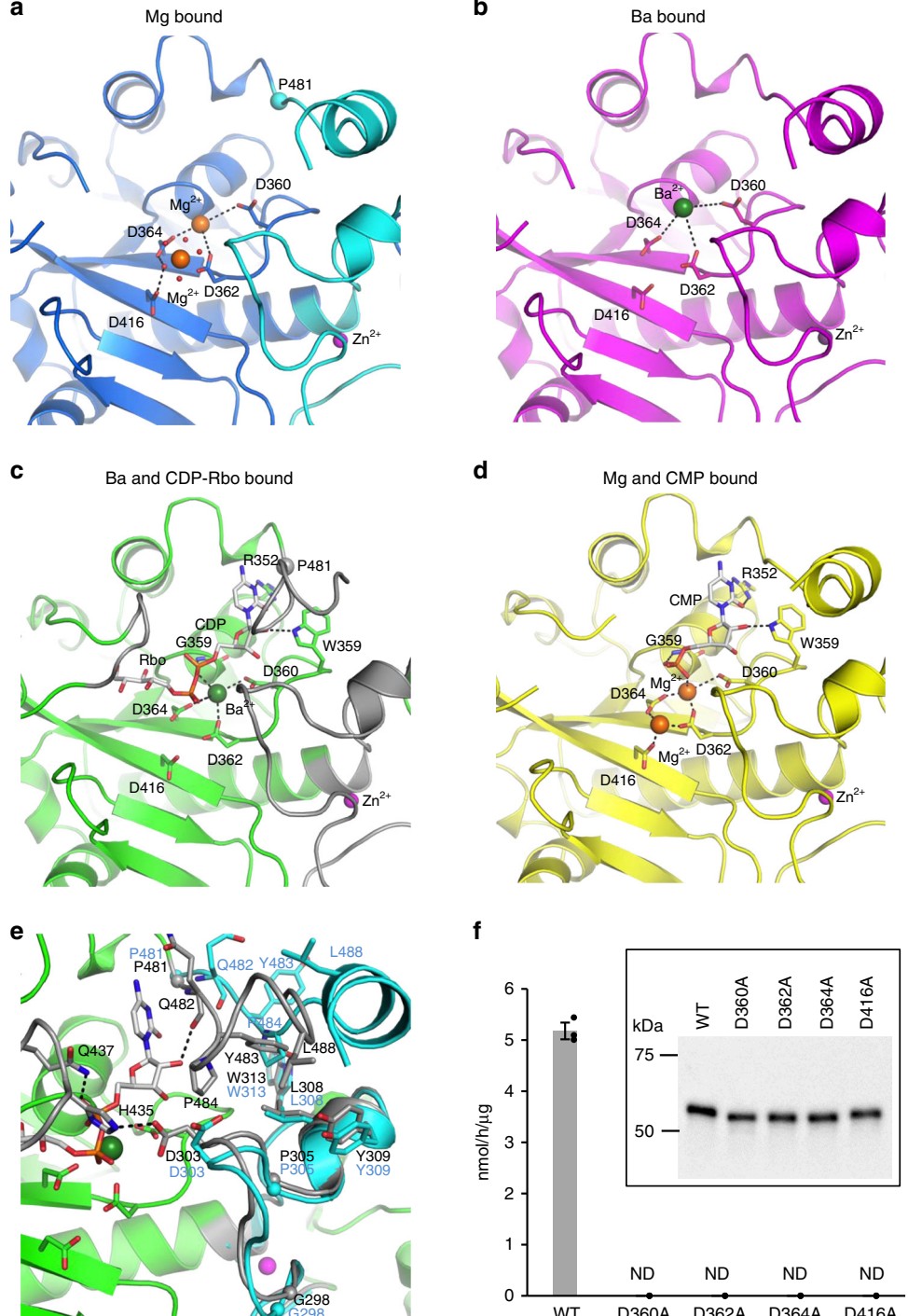

**Fig. 3 Detailed structures around the active site.** The main chain of sFKRP is shown in the cartoon model. Black dotted lines represent hydrogen bonds or ionic interactions. Each metal ion is shown by colored spheres: $Mg^{2+}$ by orange, $Zn^{2+}$ by purple, and $Ba^{2+}$ by green. **a** $Mg^{2+}$ bound structure. Water molecules which coordinate $Mg^{2+}$ at site II are shown as red dots. **b** $Ba^{2+}$ bound structure. **c** CDP-Rbo (stick model) and $Ba^{2+}$ complex structure. The regions involved in the interaction with CDP-Rbo are shown in gray. **d** CMP (stick model) and $Mg^{2+}$ complex structure. **e** Comparison with CDP-Rbo bound (corresponds to (**c**), green and gray) and substrate-free (corresponds to (**a**), sky blue) structures. The start point (P481) of conformational change induced by ligand binding is marked with a small sphere. **f** Enzymatic activities of sFKRP (WT, D360A, D362A, D364A, and D416A) with CDP-Rbo and the RboP-(phospho-)core M3 peptide. ND, not detected. Average values ± SE of three independent experiments are shown. Each dot represents one data point. Inset: immunoblot analysis of sFKRP proteins immunoprecipitated from the culture supernatant to normalize input sFKRP. Source data are provided as a Source Data file.

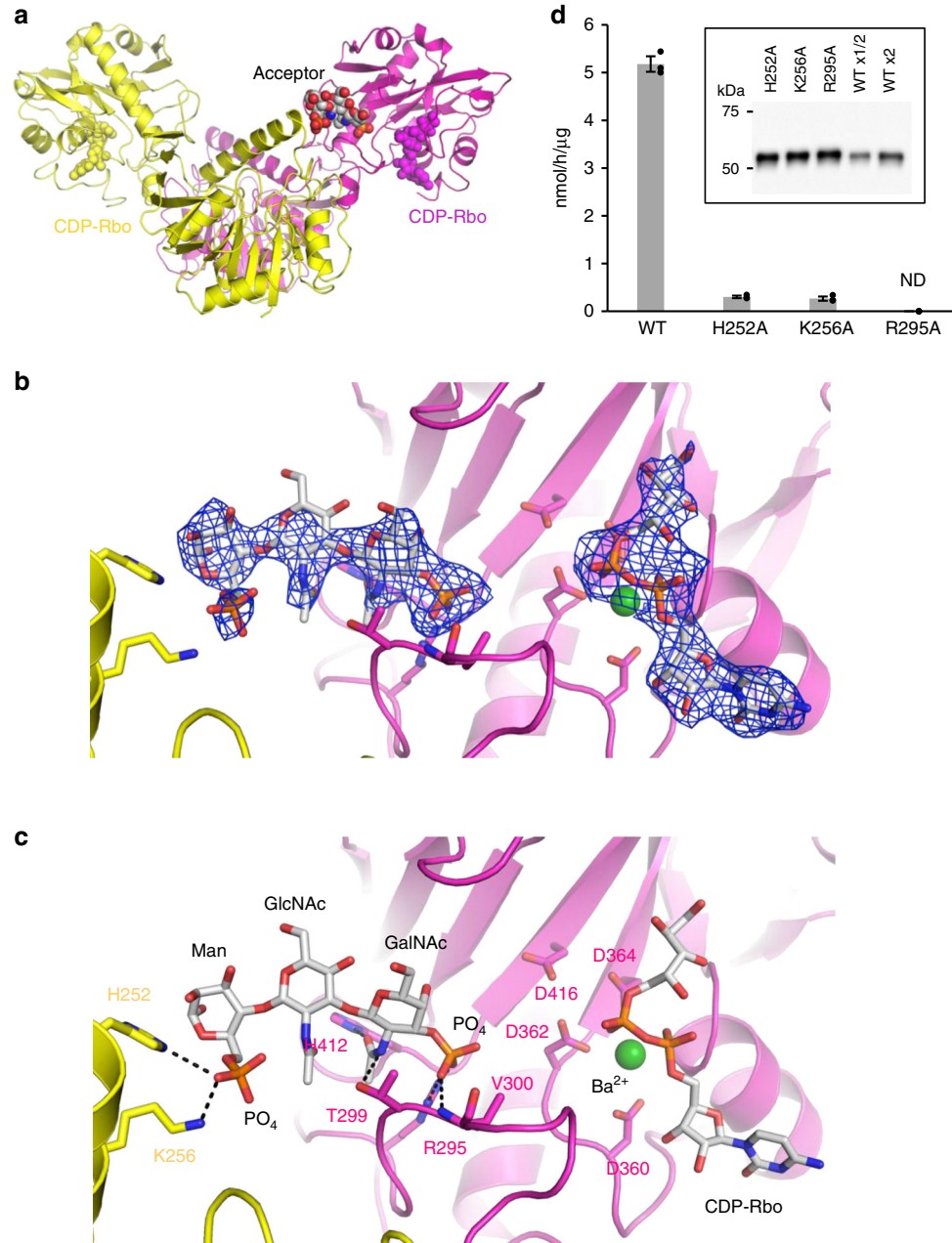

**Fig. 4 Structure of sFKRP in complex with the acceptor glycopeptide. a** Two sFKRP subunits are shown in yellow and purple, respectively. CDP-Rbo (yellow and purple) and the phospho-(phospho-)core M3 moiety (CPK color) are shown by sphere models. **b** Fo-Fc omit map around phospho-(phospho-) core M3 (left) and CDP-Rbo (right). The contour level of the map is 3.0 σ. **c** Interaction between the protomeric dimer and phospho-(phospho-)core M3 moiety. The trisaccharide is shown as a stick model and interactions between sFKRP are depicted as dotted black lines. **d** Enzymatic activities of sFKRP (WT, H252A, K256A, and R295A) with CDP-Rbo and the RboP-(phospho-)core M3 peptide. ND, not detected. Average values ± SE of three independent experiments are shown. Each dot represents one data point. Inset: immunoblot analysis of sFKRP proteins immunoprecipitated from the culture supernatant to normalize input sFKRP. Source data are provided as a Source Data file.

To confirm the observed interactions between the acceptor glycopeptide and sFKRP, mutational analyses on His252, Lys256 and Arg295 were performed (Fig. 4d). The Arg295Ala mutation lost the enzyme activity of sFKRP, indicating that the interaction between Arg295 and the phosphate group of RboP is essential for this activity. We also found that the His252Ala and Lys256Ala mutations significantly reduced the activities, suggesting the importance of the interaction between the phospho-6Man moiety of the acceptor glycopeptide and the stem domain.

In addition, we performed a mass spectrometry experiment to determine whether or not the phosphate group of the phospho-6Man moiety is required for the ligand recognition of sFKRP. For this purpose, we prepared a derivative of the acceptor glycopeptide that does not have the phosphate group at the 6th-position of O-Man (RboP-core M3 peptide) (Supplementary Fig. 5) and examined its reactivity with FKRP. As shown in Fig. 5 and Supplementary Fig. 6, only the substrate peak (S2; RboP-core M3 peptide) was detected in both the absence and presence of sFKRP, while the product peak [P1; RboP-RboP-(phospho-)core M3 peptide] was detected using its usual acceptor [S1; RboP-(phospho-)core M3 peptide]. These results indicate that the phosphate group at the 6th-position of O-Man is essential for FKRP activity.

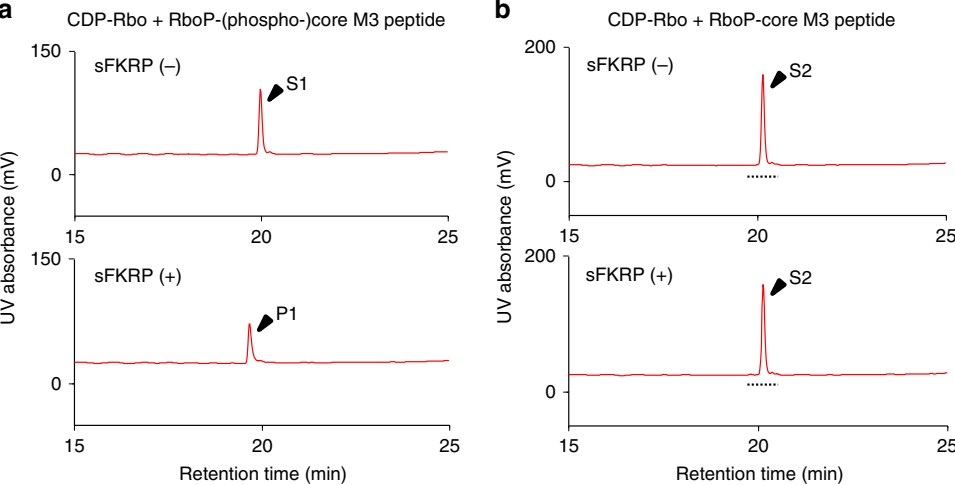

**Fig. 5 Requirement of the phosphate residue at the 6th-position of *O*-Man. a** Enzymatic activity of sFKRP with CDP-Rbo and the RboP-(phospho-)core M3 peptide. The products were analyzed by HPLC. Upper, without sFKRP; lower, with sFKRP. S1, acceptor substrate [RboP-(phospho-)core M3 peptide]; P1, product of the reaction of sFKRP with CDP-Rbo. **b** No enzymatic activity was observed for sFKRP with CDP-Rbo and the RboP-core M3 peptide. The products were analyzed by HPLC. Upper, without sFKRP; lower, with sFKRP. S2, acceptor substrate (RboP-core M3 peptide). Eluates with the retention times indicated by dotted lines were subjected to MS analysis (Supplementary Fig. 6).

On the basis of these experiments, we concluded that the two spatially separated phosphate groups of the acceptor glycopeptide interact with two distinct subunits as observed in the crystal structure. This well explains the requirement of the oligomeric structure of sFKRP for its enzymatic activity.

## Discussion

The structure of the core region of the catalytic domain is similar to that in NTase family proteins. Most known members of the NTase superfamily transfer NMP from NTP to an OH group of the acceptor molecules and release pyrophosphate (PPi) (NTP → NMP + PPi)[29]. In contrast, FKRP releases CMP from CDP-Rbo and transfers a RboP moiety to an OH group of the acceptor, RboP-(phospho-)core M3[11]. Both reactions are dehydration/condensation reactions with hydroxyl and phosphate groups, but the leaving group of the donor substrate in FKRP is different from that in typical NTase, even if FKRP has common sequence motifs, [DE]h[DE]h and h[DE]h (where h indicates a hydrophobic amino acid), and a protein fold of NTase[29] (Supplementary Figs. 1a and 2). Our structural analysis revealed a similarity between the catalytic reactions of FKRP and NTase. NTase requires two divalent metal ions, such as $Mn^{2+}$ and $Mg^{2+}$, which interact with the donor and/or acceptor molecule in the active site (Supplementary Fig. 1b, right panel)[36]. In the crystal structure of ANT(2")-Ia, a representative NTase, one metal ion at binding site I coordinates two aspartic acid residues (Asp44 and Asp46) and the diphosphate moiety of the donor molecule. The other metal ion at binding site II coordinates three aspartic acid residues (Asp44, Asp46 and Asp86) and the OH group of the acceptor molecule. Based on the results for ANT(2")-Ia and the DNA polymerase β subunit[36,37], it has been proposed that the third aspartic acid residue (Asp86) interacting with a metal ion at site II acts as an acidic base and leads to electrophilic polarization of the OH group by coordinating the metal ion. The crystal structure of the sFKRP-CMP complex revealed a similarity of the metal ion coordination sites to those of NTase; one $Mg^{2+}$ at site I coordinates Asp362 and Asp364, while the other $Mg^{2+}$ at site II coordinates Asp362, Asp364, and Asp416 (Fig. 3d and Supplementary Fig. 1b). This similarity suggests that FKRP and NTase share a catalytic mechanism.

Our crystal structure suggests that Asp416 would interact with the Rbo moiety of an acceptor glycopeptide via the metal ion at site II. However, we could not obtain direct evidence to support this; no electron density of the terminal Rbo moiety of the RboP-(phospho-)core M3 peptide was observed in the crystal structure of the sFKRP-acceptor glycopeptide complex. Within the complex, the distance between the metal-binding site II and the phosphorous atom of the RboP moiety of the acceptor molecule was sufficient to reasonably model the missing Rbo moiety. Site-directed mutagenesis studies showed that three aspartic residues (Asp362, Asp364, and Asp416) conserved in the NTase superfamily are essential for the catalytic activities of sFKRP (Fig. 3f). In addition, His42 in ANT(2")-Ia, which corresponds to Asp360 in FKRP, is an essential residue for interacting with the nucleotide donor, ATP[35,36]. This coincides well with our finding that Asp360 in FKRP was essential for the enzymatic activity and interacts with CDP-Rbo via a metal ion in site I. Moreover, a CDP-Rbo soaking experiment resulted in unintentional degradation to CMP and RboP when an $Mg^{2+}$ bound crystal was used as the parent crystal. This finding indicates that the cleavage between α phosphorous and β phosphorous of CDP-Rbo occurred in the presence of $Mg^{2+}$. These results strongly support a hypothesis that the metal ion ($Mg^{2+}$) at site II has catalytic function. Further structural and biochemical studies will be needed to account for the difference of the leaving group between FKRP and other NTase proteins.

We compared FKRP and FKTN, which show different substrate specificities, by their primary sequences. The amino acid sequence identity between their stem domains (45–287 in FKRP and 44–248 in FKTN) was very low (8.2%), suggesting that the molecular mechanism, particularly the recognition mechanism of the acceptor molecule, would be different. On the other hand, the sequence identity at the catalytic domains was 20% (288–451 in FKRP and 249–461 in FKTN). Thus, the structure of the catalytic domain of FKTN would be similar with that of FKRP; the structural similarity of their catalytic domains was also suggested by comprehensive sequence analysis[29]. We therefore made a homology model of the catalytic domain of FKTN based on the FKRP structure by using a Swiss-model (Supplementary Fig. 9)[38]. The model showed that FKTN has no zinc finger loop and Arg residue corresponding to Arg295 in FKRP, while the three Asp

residues corresponding to Asp362, Asp364 and Asp416 in FKRP are conserved. These facts suggest that the sugar acceptor recognition (i.e., core M3 recognition) is completely different between FKRP and FKTN, although the metal and sugar donor recognitions were similar to each other.

While it was reported that FKRP formed a dimer in vivo through an intermolecular disulfide bond at Cys6 in the cytoplasmic region[26], our crystal structure and SEC-SAXS analysis showed that the lumenal side of FKRP forms a tetramer with tight interactions. Our results agree with those of a previous study in which FKRP formed oligomers[26]. Further, we have revealed a functional implication for the FKRP oligomerization on the basis of structure-based mutational analysis. Since recognition of the acceptor glycopeptide requires interactions by two distinct subunits in the protomeric dimer, the dimeric form of sFKRP is essential for its enzymatic activity. We showed the importance of the protomeric dimer using crystal structures and SEC/enzymatic measurements of the mutants, but the significance of the tetramer remains elusive.

Our structural studies clarified a unique ligand recognition mechanism of FKRP. The dissimilarity of FKRP against NTase family members is characterized by the zinc finger loop and the stem domain (Supplementary Fig. 1). These two regions cover the active site of the catalytic domain. Arg295, which is in the zinc finger loop, is essential for the enzyme activity and interacts with a phosphate of the RboP residue of the acceptor glycopeptide (Fig. 4c, d). Another loop region interacts with the Rbo moiety of CDP-Rbo and is important for stabilization of the donor molecule bound in the active site. These two loops are conserved in only FKRP orthologues. In addition to the two loops, we found that the stem domain is also important for the acceptor recognition. Our structural and functional studies revealed that electrostatic interaction between the phosphate residue of the O-Man moiety and two amino acid residues (His252 and Lys256) in the stem domain are important for the enzymatic activity of FKRP, suggesting that the stem domain plays dual roles, functioning not only for oligomerization but also for the acceptor recognition (Supplementary Fig. 10). To date, various disease-causing mutations have been found in FKRP[18–20], and the functional characterizations of several mutations were reported[21–23]. According to our results, genetic variants at Thr293 or Val300, which make up the zinc finger loop, are associated with LGMD2I[18]. In addition, it is implicated that a specific region between residues Val300 and Ala321, which overlaps with the zinc finger loop, is important for the enzymatic activity of FKRP[23].

In this study, we demonstrated a molecular mechanism connecting dystroglycanopathy with a deficiency of the oligomerization of FKRP. Further, we revealed the substrate recognition mechanism by dimer FKRP: one end of the acceptor glycopeptide is recognized by the catalytic domain of one subunit, and the other side by the stem domain of another subunit. Finally, this study opens a path to a better understanding of the mutated FKRP causing dystroglycanopathies and should contribute to the future development of therapeutic strategies.

## Methods

**Materials**. CDP-Rbo was synthesized chemically[11]. The RboP-(phospho-)core M3 peptide was made from the phospho-core M3 peptide using a soluble form of FKTN (sFKTN). Briefly, the recombinant sFKTN was expressed in HEK293T cells and immunoprecipitated from the culture supernatants with the anti-c-Myc antibody agarose conjugate (rabbit polyclonal; Sigma). sFKTN bound to agarose was used as the enzyme source. The enzyme reactions were performed in a 500 μL reaction volume at 310 K for 24 h using the following buffer conditions: 100 mM MES (pH 6.5), 1 mM CDP-Rbo, 0.2 mM phospho-core M3 peptide, 10 mM MnCl$_2$, 10 mM MgCl$_2$, 0.5% Triton X-100, and 220 μL sFKTN-bound agarose. The reaction product was separated by reversed-phase HPLC with a Mightysil RP-18GP Aqua column (4.6 × 250 mm) (Kanto Chemical). The column was equilibrated with solvent A [0.085% trifluoroacetic acid (TFA) in distilled water], and the product

peptide was eluted at a flow rate of 1 mL/min using a linear gradient of 0–40% solvent B (0.085% TFA in acetonitrile). The RboP-core M3 peptide was obtained from the RboP-(phospho-)core M3 peptide by dephosphorylation of the phosphate group at the 6th-position of O-Man. Briefly, 10 nmol of the RboP-(phospho-)core M3 peptide was incubated with 25 U Anza alkaline phosphate (Invitrogen) at 310 K for 20 h, and the reaction product was separated by reverse-phase HPLC as described above. The production of RboP-(phospho-)core M3 peptide and RboP-core M3 peptide was confirmed by MALDI-TOF-MS[39]. Briefly, the samples were desalted using GL-Tip SDB (GL Sciences), and mixed with an equal volume of matrix solution (2,5-dihydrobenzoic acid in 50% acetonitrile containing 0.1% TFA). Then, 2 μL of this mixture was dropped onto a μFocus MALDI plate (700 μm; Hudson Surface Technology) and left at room temperature to dry. MS analysis was performed on an AB SCIEX TOF/TOF 5800 system (AB SCIEX). For each spot, MS spectra were acquired in positive ion mode between m/z 800 and 4000 and accumulated from 4000 laser shots in a random raster.

**Generation of sFKRP mutants**. The expression vector for the secreted type of FKRP was prepared as follows. The PCR product encoding sFKRP (Asn33 to the C-terminus) was inserted into the EcoRI and XbaI sites of ss1-His/Myc-pcDNA3.1[11]. Each mutation was introduced by the quick change PCR method (Agilent Technologies) by using the listed primers (Supplementary Table 4).

**Protein expression and purification**. The lumenal region of human FKRP (sFKRP, residue range 45 to C-terminus) was expressed as a fusion protein with protein A at the N-terminus. sFKRP was inserted into the FseI site of pPA-IRES vector (kindly provided by Dr. James M. Rini of the University of Toronto[40]) using the primers listed in Supplementary Table 4 and an In-Fusion HD cloning kit (Takara Bio). The plasmid was transfected to HEK293S GnT- (ATCC CRL-3022) to produce a stable cell line expressing sFKRP. The stable cell line was inoculated in roller bottles for 2 or 3 weeks. The medium was harvested and then precipitated by adding 55% saturation of ammonium sulfate. The precipitant was solubilized by PBS + 0.05% NP-40 and supplemented with 5 mL of IgG Sepharose 6 Fast Flow affinity resin (GE Healthcare), that was equilibrated beforehand with PBS + 0.05% NP-40. The mixture was incubated for 1 h at 297 K and then washed with a 20 column volume of PBS + 0.05% NP-40. TEV protease was added and incubated overnight at 297 K. After cleavage of the protein tag, sFKRP was eluted by PBS + 0.01% NP-40 and then concentrated by an Amicon Ultra centrifugal filter (MWCO = 50 kDa; Merck). The sample was applied to a MonoQ (GE Healthcare) anion exchange column equilibrated with 25 mM Tris-HCl, pH 8.0, 50 mM NaCl, and 1 mM DTT. sFKRP was eluted by a linear gradient of NaCl from 50 mM to 500 mM in the same buffer. After that, sugar chains of sFKRP were removed by Endo H glycosidase (Endo Hf; New England Biolabs) overnight at room temperature, followed by amylose resin purification. Finally, the sample was applied to a Superose 6 (GE Healthcare) gel filtration column equilibrated with 10 mM HEPES-NaOH (pH 7.5), 150 mM NaCl, and 1 mM DTT. The peak fractions were collected and concentrated up to ~7.5 mg/mL with an Amicon Ultra (MWCO = 50 kDa; Merck).

**Crystallization and structure determination**. Purified sFKRP was subjected to initial crystallization screening using an automated protein crystallization and monitoring system[41]. We used crystal screening kits, Wizard Classic 1 and 2 (RIGAKU), Index (Hampton Research), and Protein Complex (QIAGEN), for initial screening at a protein concentration of 3 mg/mL. Subsequent optimization of crystallization conditions was carried out by the hanging-drop vapor diffusion method by mixing equal volumes of protein (3 mg/mL) and reservoir solutions. Crystals of the apo form of sFKRP were obtained under two different reservoir conditions at 293 K: a Mg$^{2+}$ containing condition [4–8% PEG 4000, 0.1 M HEPES-NaOH (pH7.5) and 0.2 M MgCl$_2$] and a Ba$^{2+}$ containing condition [4–8% PEG 4000, 0.1 M HEPES-NaOH (pH7.5), 0.5 M LiCl$_2$ and 20 mM BaCl$_2$]. To improve the crystal quality, we added seed crystals just after making the crystallization drops. Before the X-ray diffraction experiments, the crystals were soaked in reservoir solution supplemented with 30% ethylene glycol as a cryo-protectant. The diffraction data set was collected at 100 K using an X-ray wavelength of 1.9000 Å at BL-17A of the Photon Factory (Tsukuba, Japan) and processed and scaled using XDS[42] and Aimless from the CCP4 software package[43], respectively. Initial phases were determined by Ba-SAD (single wavelength anomalous dispersion) method using Auto-SHARP[44]. Ligand complex structures were obtained by the soaking method. 5 mM CDP-Rbo, 5 mM CMP and 10 mM acceptor peptide were used for soaking experiments, respectively. sFKRP crystals were transferred to 1 μL of reservoir solution containing each ligand, and incubated 2 h at 293 K. Structure refinement was performed by Coot[45] and phenix.refine[46]. Data collection and refinement statistics are summarized in Supplementary Table 1. The Mg$^{2+}$ bound form was determined by the molecular replacement method using the Ba$^{2+}$ bound form as a search model. The crystal structures of the Ba$^{2+}$ and Mg$^{2+}$ bound forms were refined at 2.25 Å and 2.06 Å resolution, respectively. The crystal structures of the two crystal forms were almost identical, with a root-mean-square deviation (RMSD) of 0.237 Å (1,751 Cα atoms). Other complex structures were determined by the molecular replacement method using the Mg$^{2+}$ bound form as a search model.

The structural similarity between the stem domain of FKRP and ppGalNAcT-10 was identified by Dali pairwise comparison[47], providing a Z-score of 11.9 and sequence identity of 10%. The Z-score and RMSD between sFKRP and ANT(2")-Ia (PDB ID: 5KQJ) of each catalytic domain are 10.2 and 3.4 Å, as determined by Dali pairwise comparison analysis[48] (Supplementary Fig. 1a).

**SEC-SAXS analysis.** The SEC-SAXS experiments were performed as follows. A WTC-030S5 column (Wyatt Technology) connected to an Alliance HPLC system (Waters) was used for SEC. The scattering data for sFKRP were collected at 298 K using BL-10C at the Photon Factory (Tsukuba, Japan), with a wavelength of 1.0000 Å and detector length of 2.0113 m using a Pilatus 2 M detector (Dectris). The sample was applied the column, equilibrated with the buffer [20 mM HEPES-NaOH (pH 7.5), 200 mM NaCl, and 2 mM $MgCl_2$], at a flow rate of 0.6 mL/min for the first 6 min and at 0.05 mL/min for the elution peak. The scattering images were obtained for 20 s per frame at a flow rate of 0.05 mL/min. All scattering images were processed using the programs SAngler[49] and PRIMUS[50].

**Oligomerization analysis of sFKRP mutants.** Since the mutant proteins of Tyr88Phe, Ser221Arg, and Lue276Ile could not be purified, these oligomerization states were analyzed as follows. Each expression vector for secreted type of sFKRP was transfected into HEK293T with polyethyleneimine, and expressed for 48 h at 310 K under an atmosphere of 5% $CO_2$. The medium was harvested and applied to a Superdex 200 increase 5/150 column (GE Healthcare) after filtration through a 0.22 μm PVDF membrane. Elution fractions were collected and subjected to Western blotting analysis. To detect the sFKRP protein, c-Myc tag antibody [c-Myc antibody (9E10): sc-40; Santa Cruz Biotechnology] was used.

**Enzymatic assay for sFKRP.** The secreted type of WT or mutant sFKRP was expressed in HEK293T cells using Lipofectamine 3000 (Life Technologies Japan). The expressed proteins were immunoprecipitated from the culture supernatant with anti-c-Myc antibody-agarose (rabbit polyclonal; Sigma). The proteins bound to the agarose were used as the enzyme sources. sFKRP expression levels were determined by SDS-PAGE (7.5% acrylamide) followed by Western blotting[11]. FKRP enzymatic reactions were performed in 25 μL of reaction buffer (100 mM MES-NaOH, pH 6.5, 0.5% Triton X-100) containing 10 mM $MnCl_2$, 10 mM $MgCl_2$, 200 μM CDP-Rbo, 40 μM acceptor peptide [RboP-(phospho-) core M3 peptide] and the enzyme-bound agarose (384.6 ng protein) at 310 K for 15 min, except where otherwise indicated. Alternatively, because the Ser221Arg and Leu276Ile mutants were scarcely secreted to the culture media, cell lysates were prepared using reaction buffer supplemented with 50 mM Tris-HCl (pH 7.4), 500 mM NaCl, 10 mM $MnCl_2$, 10 mM $MgCl_2$ and protease inhibitor cocktail (Halt; Thermo Fisher, Waltham, MA). The lysates (31.6 μg total protein) were used as the enzyme sources, and FKRP enzymatic reactions were performed at 310 K for 2 h. The enzymatic activities of the recombinant sFKRP were determined by subtracting the FKRP activity of the cell lysate without an expression plasmid. For the assays to examine the divalent cation requirements, reaction buffers containing 10 mM EDTA, $MgCl_2$, $MnCl_2$, $CaCl_2$ or $BaCl_2$ were used. For the assays to investigate the requirement of the phosphate group at the 6th-position of O-Man, the acceptor peptide with or without phosphorylation at the 6th-position of O-Man was used and the enzymatic reactions were performed at 310 K for 1 h. Each product was analyzed by reversed-phase HPLC and MS as described above. Each enzymatic activity was calculated from the product peak area.

**Reporting summary.** Further information on research design is available in the Nature Research Reporting Summary linked to this article.

## Data availability

The coordinates and structure factors for sFKRP have been deposited in the Protein Data Bank (PDB) under the accession codes 6KAJ for $Ba^{2+}$ and CDP-Rbo complex, 6KAK for $Mg^{2+}$ bound form, 6KAL for $Mg^{2+}$ and CMP complex, 6KAM for $Ba^{2+}$, CDP-Rbo and acceptor complex, 6KAN for $Ba^{2+}$ bound form, 6L7S for $Mg^{2+}$ bound form (Zn peak data), 6L7T for $Mg^{2+}$ bound form (Zn low remote data), and 6L7U for Ba-SAD data. The source data underlying Figs. 2c–e, 3f, 4d and Supplementary Fig. 4 are provided as a Source Data file. Other data are available from the corresponding authors upon reasonable request.

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

## Acknowledgements

We thank Drs. N. Shimizu, S. Saijo, and N. Suzuki (KEK/SBRC) for SAXS experiments; and beamline staffs of Photon factory (Tsukuba, Japan) and NSRRC (Shinchu, Taiwan) for X-ray crystallographic analysis; and Mr. T. Ohta (KEK/SBRC) for technical assistance of protein expression and purification; Dr. J. M. Rini (Univ. of Toronto) for providing us an expression vector, pPA-IRES. This work was supported by PDIS and Platform Project for Supporting Drug Discovery and Life Science Research (Basis for Supporting Innovative Drug Discovery and Life Science Research, BINDS) from AMED (JP18am0101083 and JP18am0101071); the National Center of Neurology and Psychiatry (NCNP) Intramural Research Grant 29-4 (to T. Toda and T.E.); AMED grant numbers JP18gm0810010, JP18ek0109197 (to H.M. and M.K.), and 19ek0109249 (to T. Toda); the Japan Society for the Promotion of Science Grants (JSPS) KAKENHI JP16K07284 (to N. K.), JP26840029 (to N.K.), JP17H03987 (to H.M.), JP17H01563 (to T. Toda), JP19H05648 (to T.E, K.K., and R.K.), and JP16K08262 (to T.E.); the Ministry of Education, Culture, Sports, Science and Technology of Japan Grant JP17919697 (to M.K.); Takeda Science Foundation Grant (to M.K.).

## Author contributions

N.K., H.M., T.E. and R.K. designed the project; N.K., R.I., H.M., H.T. M.K., K.K. and T. Toda performed the experiments; T. Tanaka and M.M. synthesized the glycosylated peptides; N.K., R.I., H.M. and T.S. analyzed the data; N.K., R.I., H.M., T.S., T.E. and R.K. wrote the paper.

## Competing interests

The authors declare no competing interests.
