## [Peer Review File · Nature Communications]

Reviewers' comments:

Reviewer #1 (Remarks to the Author):

The Authors have tackled a compelling and important subject. Their crystallographic data are novel and interesting and could represent a real breakthrough in the field of muscular dystrophies related to FKRP at large, and specifically in our understanding of such diseases at the molecular level. I have a series of comments that I believe should be addressed before considering this manuscript suitable for publication.

_I suggest that the title more specifically alludes at the crystallographic nature of this work. A possible suggestion being "High-resolution crystal structures of fukutin-related protein (FKRP), a ribitol-phosphate transferase related to muscular dystrophy. Insights into its oligomerization and ligand recognition.

_ISPD and the work published by Gerin et al. (2016) on Nature Communications should be also mentioned in the Introduction. Esapa et al., 2002 (Hum Mol Genet), de Paula et al. 2003 (Eur J Hum Genet), or others available papers (see also the recent Henriques et al., 2019, Hum Mut), should be also quoted because of the previous investigation of mutated FKRP or for reporting several missense mutations causing phenotypes. For example, some of the mutations reported by the dePaula et al are very near to the enzymatically crucial Arg295 (in the zinc finger loop), as are Thr 293 or Val 300.

_Introduction (bottom of page 3). "The synthesized phospho-core M3 structure..." should be changed to "The dystroglycan core protein modified with the synthesized phospho-core M3 structure is transported to the Golgi apparatus"

_Line 1 p.5 (Results), please specify there "crystal structure of human FKRP". Over the following line, please also specify the total number of residues of the recombinant protein (451 aa I reckon).

_Please state more clearly that the major interface contributing to the formation of the dimers seem to be the one formed by the reciprocal stem domains. Correct?

_Page 7 please re-phrase "revealed breakdowns in their tetrameric structures". What does it mean exactly? For example, L276I seems to be present also at a higher oligomerization state (Fig.2C). Do at least some of the mutants analyzed still form homodimers? Possibly, the estimated molecular weights should be reported not only for the WT (214 KDa) but as well for the three analyzed mutants.

_How do the glycosyltransferase activities (reported as % of the WT) detected for the Tyr88Phe, Ser221Arg and Leu276Ile mutants relate with the phenotype severity observed in patients?

_I agree that the experiments with different divalent cations on the recombinant protein suggest that Mg²⁺ or perhaps Mn²⁺? (see next point as well) are the metal ions present also in the native enzyme. However, until this could be ultimately proven by some analysis carried out on the native protein purified from cells/tissues I guess, it should be stated clearly that the above would be the most likely suggestion(s) arising from the analysis of the recombinant protein(s).

_Fig.S4, the effect induced by Mn²⁺ seems much larger than those observed with all the others divalent cations. Was any Mn²⁺ present in the buffer used for solubilizing the recombinant protein, was it added to some of the crystallization conditions or added to the crystals? I did not find any specific information on Mn²⁺ in Methods apparently. Do the Authors imply that Mn²⁺ would have the same coordination shells than Mg²⁺? In the Legend to Figure S4: please add "respectively" after 10 mM EDTA, MgCl₂, MnCl₂, CaCl₂ or BaCl₂.

_Figure 3: based on similarities with canonical NTases, the Authors state that CDP-Rbo is likely to interact with the metal in site II but no metal was detected there I reckon. Was any Ba²⁺ observed in site II?

_Upon catalysis, the CDP-Rbo donor is transformed into CMP and the ribitol-P transferred onto dystroglycan. The CMP-bound structure with Mg²⁺ has been solved, but not with Mn²⁺. However, what is happening next? Is CMP released when a new CDP-Rbo is available to enter the catalytic site? Is it implied that site I is not completely occupied by a metal atom? Again, is Mg²⁺ or Mn²⁺ the metal found in native conditions? In general, an overall synoptic cartoon indicating the two domains (stem and catalytic), the relevant areas (site I and site II) within the catalytic domain in the substrate-free state, the subsequent reaction steps involving CDP-Rbo (donor) and RboP- (phospho-)core M3 (acceptor) as well as the bidimensional chemical structures of donor and acceptor, and possibly highlighting some of the points still open (Mg vs Mn?), could be helpful especially for the general audience and could be proposed within the Discussion.

_Figure 3F: Why was Asp360 not mutated to Ala too? It should be involved in coordination at site I (weaker interactions with Mg²⁺). Indeed, as reported in Fig. S2, in the fly an alanine residue occupies the same topological position.

_“The CDP-Rbo binding evokes conformational changes” paragraph is written in a sketchy fashion. For example, “These changes were related to each other and interactions were found among them” (interactions between changes?). Is the CDP-Rbo unbound form simply the substrate-free form? Please use there and elsewhere “substrate-free” instead of “unbound”. “The C-terminal fragment in subunit A adopts an alpha helix”, please re-phrase. Were all the reported conformational changes (Fig. 3F) identified comparing the CDP-Rbo bound form with Ba²⁺ to the substrate-free form with Ba²⁺, or also with Mg²⁺? In case, did any relevant discrepancies emerge between the substrate-free Ba²⁺ and Mg²⁺ forms, aside from the coordination sites?

_Page 10: please amend to “In addition, we performed a mass spectrometry experiment to determine...”.

Reviewer #2 (Remarks to the Author):

The dystroglycan (DG) complex plays a myriad of important physiological functions, ranging from muscle development to viral infection. Many of its function depend on the proper glycosylation of alpha-DG, which requires the concerted effort of more than 10 different enzymes. Fukutin and fukutin-related protein (FKRP) have been identified recently as ribitol-phosphate (RboP) transferases that are involved in the biosynthesis of the core M3 glycan of alpha-DG. Mutations of either Fukutin or FKRP can lead to dystroglycanopathy, highlighting the functional significance of these enzymes. Notably, these two enzymes represent the only two known examples of RboP transferases in human cells. In this manuscript, Kuwabara et al. have determined a series of crystal structures of FKRP at different functional states. This study is of interest, and represents an advance in our understanding of this unique family of enzymes. However, the draft in its current state raises several issues that are outlined below, which need to be addressed before it can be published.

1. The authors need to clarify the functional importance of FKRP tetramer. Based on the structural and biochemical analyses, the “protomeric dimer” of FKRP is clearly important. However, the functional significance of the tetramer is unclear. However, the authors claim in several places that the FKRP “oligomer” is important, which is vague and not accurate. The authors should either state that the dimer is important, but the significance of the tetramer is unclear, or set out to evaluate the importance of the tetramer by making mutations in the tetramer interface. Does FKRP exist as a dimer or higher-order oligomer *in vivo*?

2. The authors need to clarify the functional importance of site I and site II metal ions.

1) Figure 3B: can the authors show the anomalous difference map for Ba, like the one they showed in Figure 1C for Zn? Can the authors comment on why only one Ba atom is found in this structure, and why it is seen at the weak binding site I, but not the stronger site II?

2) Figure 3C and 3D: based on these two structures, the authors suggest that “the metal ion at site II, which could not be observed in the crystal structure, also interacts with the donor molecule CDP-Rbo” (Page 8). This is not very clear in the current presentation. Can the authors overlay the

two structures, and describe the distances of site II metal to CDP-Rbo atoms if it is present?

3) Figure 3F: Asp362 and Asp364 are involved in binding to both site I and site II metal ions, so their mutations could affect metal binding to both sites. Have the authors made the D360A mutant and tested its activity?

4) In the Discussion section, the authors suggest that FKRP has a similar catalytic mechanism as NTase, with the site II metal playing a catalytic function. This section needs to be consolidated, and some figures are necessary to illustrate the active site comparison between FKRP and NTase.

3. The structure with the RboP-(phosphor-)coare M3 peptide bound is the most important result in this paper. It is unfortunate that the electron density for several parts of the acceptor molecule are unclear, especially for the Rbo moiety. How do the authors know that an intact acceptor glycopeptide, rather than a degradation product of this molecule, is present in the crystal? Have the authors tried to use a catalytically dead mutant of FKRP to obtain the complex structure?

4. The presentation of Figure 4 is also confusing and needs to be thoroughly edited.

1) Figure 4A: the authors should label the CDP-Rbo and the phosphor-core M3. Also, the CDP-rbo appears all purple here, not "yellow and purple" as described in the legend, which is confusing.

2) Figure 4B: the authors should consider showing the Fo-Fc map for both the CDP-Rbo and the phosphor-core M3 molecules, since this is a new (and the most important) structure.

3) Figure 4C: The interaction between H252/K256 and phosphor-6Man should be indicated with dashed lines. Also, are the labels for GalNAc and GlcNAc swapped? "RboP-3GalNAc β 1-3GlcNAc β 1-4(phospho-6)Man α 1" is described in the text (Page 9), so GlcNAc should be in the middle. Also related to this, "the N-acetyl group of the GlcNAc moiety interacts with Thr299 and His412" (Page 9)—should be the N-acetyl group of GalNAc instead? This sentence is also not in the correct place—both the sentence before it and after it are describing the phosphate group of RboP.

5. Based on the structure of FKRP, can the authors make a structural model for Fukutin, and comment on their similarities and differences, especially in the regions that bind to the acceptor molecules? This would shed light on their different substrate specificity.

Point-by-Point Responses to the Referee Comments/Suggestions

Comments of Referee 1

I suggest that the title more specifically alludes at the crystallographic nature of this work. A possible suggestion being “High-resolution crystal structures of fukutin-related protein (FKRP), a ribitol-phosphate transferase related to muscular dystrophy. Insights into its oligomerization and ligand recognition.

Response: Thank you for the suggestion. We changed the title of the manuscript to “Crystal structures of fukutin-related protein (FKRP), a ribitol-phosphate transferase related to muscular dystrophy: insights into its oligomerization and ligand recognition”.

ISPD and the work published by Gerin et al. (2016) on Nature Communications should be also mentioned in the Introduction.

Response: Thank you for the comment. We revised the Introduction section (page 4 lines 21-23) as follows to mention ISPD and to provide reference [25]: “FKTN transfers the first RboP to the 3rd-position of GalNAc from CDP-Rbo, which is synthesized from RboP and CTP by isoprenoid synthase domain-containing (ISPD) [11,25]”

25. Gerin et al. (2016) *Nat Commun* 7:11534

Esapa et al., 2002 (Hum Mol Genet), de Paula et al. 2003 (Eur J Hum Genet), or others available papers (see also the recent Henriques et al., 2019, Hum Mut), should be also quoted because of the previous investigation of mutated FKRP or for reporting several missense mutations causing phenotypes. For example, some of the mutations reported by the dePaula et al are very near to the enzymatically crucial Arg295 (in the zinc finger loop), as are Thr 293 or Val 300.

Response: Thank you for the suggestion. Since we cited only a review article [17] in the original manuscript, we added new references as suggested [18-23] (page 4 line 7). We also added the following passage to the Discussion to discuss the FKRP patient mutations and mutant analysis (page 13 line 2 from the bottom - page 14 line 5):

“To date, various disease-causing mutations have been found in FKRP [18-20], and the functional characterizations of several mutations were reported [21-23]. According to our results, genetic variants at Thr293 or Val300, which make up the zinc finger loop, are associated with LGMD2I [18]. In addition, it is implicated that a specific region between residues Val300 and Ala321, which overlaps with the zinc finger loop, is important for the enzymatic activity of FKRP [23].”

18. de Paula F et al. (2003) *Eur J Hum Genet* 11:923-930

19. Beltran-Valero de Bernabé D et al. (2004) *J Med Genet* 41:e61
20. Brockington M et al. (2001) *Am J Hum Genet* 69:1198-1209
21. Esapa CT et al. (2002) *Hum Mol Genet* 11:3319-3331
22. Esapa CT et al. (2005) *Hum Mol Genet* 14:295-305
23. Henriques SF et al. (2019) *Hum Mutat* 40:1874-1885

Introduction (bottom of page 3). “The synthesized phospho-core M3 structure...” should be changed to “The dystroglycan core protein modified with the synthesized phospho-core M3 structure is transported to the Golgi apparatus”

Response: Thank you for the comment. We changed the sentence as suggested (page 3, line 6 from the bottom): “The dystroglycan modified with the synthesized phospho-core M3 is transported to the Golgi apparatus.”

Line 1 p.5 (Results), please specify there “crystal structure of human FKR P”. Over the following line, please also specify the total number of residues of the recombinant protein (451 aa I reckon).

Response: Thank you for the comment. We changed the sentences as follows (page 5, lines 4-6): “Initially, we determined the crystal structure of human FKR P without substrates. A soluble form of FKR P (sFKR P: residues 45 to C-terminus of FKR P, 451 amino acid residues in total) was purified and crystallized. . .”

Please state more clearly that the major interface contributing to the formation of the dimers seem to be the one formed by the reciprocal stem domains. Correct?

Response: Thank you for the suggestion. We revised the manuscript as follows to clarify that the main interaction of the dimer is formed by the stem domains (page 5, bottom of the first paragraph). The value of the buried surface area was corrected to 1,642 Å² from 1,652 Å² by re-calculation.

“The two protomeric dimers are related by a two-fold axis in the tetramer with a buried surface area of 1,642 Å². For the dimer-dimer interface, the contribution of the two stem domains is largest (730 Å²), followed by the contribution of the stem and catalytic domains (504 Å²), and finally that of the two catalytic domains (336 Å²).”

Page 7 please re-phrase “revealed breakdowns in their tetrameric structures”. What does it mean exactly? For example, L276I seems to be present also at a higher oligomerization state (Fig.2C). Do at least some of the mutants analyzed still form homodimers? Possibly, the estimated molecular weights should be reported not only for

the WT (214 kDa) but as well for the three analyzed mutants.

Response: Thank you for the important comments. As you point out, high molecular weight bands were clearly observed in Leu276Ile (Figure 2C). Corresponding bands were also observed in the WT and other mutants. Gel filtration analysis showed that the high molecular weight species of the WT were eluted at the void volume, which theoretically corresponds to 1,300 kDa (Superdex200 increase, GE), suggesting that they are aggregates of sFKRP. Thus, recombinant sFKRP and its mutants seem to be easily aggregated in solution.

The molecular weight of the WT sFKRP could be determined as 214 kDa using SEC-SAXS (Figure 2B). However, since the mutant proteins of Tyr88Phe, Ser221Arg, and Lue276Ile were unstable, we could not purify them. Therefore, it was impossible to analyze their molecular weights using SEC-SAXS. We were only able to estimate their molecular weights by using the gel filtration with Western blotting (Figure 2C). This method suggested that the molecular weights of the mutants were smaller than that of the WT.

Following the suggestion of the referee, we changed the sentence that included the phrase “revealed breakdowns in their tetrameric structures” as follows (page 7, lines 9-11):

“In the SEC analysis, the apparent molecular weight of the mutated proteins was lower than that of the WT, suggesting dissociation of a subunit(s) from the tetramer (Figure 2C).”

In addition, we added a sentence to the Methods section explaining that the mutant proteins of Tyr88Phe, Ser221Arg, and Lue276Ile could not be purified (page 20, lines 12-13):

“Since the mutant proteins of Tyr88Phe, Ser221Arg, and Lue276Ile could not be purified, these oligomerization states were analyzed as follows.”

How do the glycosyltransferase activities (reported as % of the WT) detected for the Tyr88Phe, Ser221Arg and Leu276Ile mutants relate with the phenotype severity observed in patients?

Response: The enzymatic activity of Ser221Arg was 5% of that of the WT, and Ser221Arg is associated with a severe form of congenital muscular dystrophy (MDC1C) [33]. The enzymatic activities of Tyr88Phe and Leu276Ile were 20% and 50% of the WT activity, respectively, and Tyr88Phe and Leu276Ile are associated with the milder limb-girdle muscular dystrophy (LGMD2I) [31,34]. In addition, the patient with Tyr88Phe mutation developed the disease at two years of age, while the patients with

Leu276Ile mutation developed the disease in the second decade of life or later. These facts suggest that the enzymatic activities of these mutants correlate with the phenotype severity observed in patients. We added the following sentence (page 7, lines 14-15) and a new reference [34]:

“The activities of these mutants seem to correlate with the phenotype severity observed in patients [31,33,34].”

34. Mercuri E et al. (2003) *Ann Neurol* 53:537-542

I agree that the experiments with different divalent cations on the recombinant protein suggest that Mg²⁺ or perhaps Mn²⁺? (see next point as well) are the metal ions present also in the native enzyme. However, until this could be ultimately proven by some analysis carried out on the native protein purified from cells/tissues I guess, it should be stated clearly that the above would be the most likely suggestion(s) arising from the analysis of the recombinant protein(s).

Response: Thank you for the important comments. While recombinant FKRP is active with Mg²⁺ or Mn²⁺ (Figure S4), it is unclear whether the native FKRP protein in cells binds Mg²⁺ or other metal ions as the referee commented. Therefore, we revised the main text as follows (page 8, lines 10-13):

“While sFKRP is active with Mg²⁺ or Mn²⁺ *in vitro*, we did not have experimental data for the metal ions of FKRP in cells. Since we could not obtain crystals of the Mn²⁺-bound form, we used the Mg²⁺-bound form for structural and biochemical analysis.”

Fig.S4, the effect induced by Mn²⁺ seems much larger than those observed with all the others divalent cations. Was any Mn²⁺ present in the buffer used for solubilizing the recombinant protein, was it added to some of the crystallization conditions or added to the crystals? I did not find any specific information on Mn²⁺ in Methods apparently. Do the Authors imply that Mn²⁺ would have the same coordination shells than Mg²⁺? In the Legend to Figure S4: please add “respectively” after 10 mM EDTA, MgCl₂, MnCl₂, CaCl₂ or BaCl₂.

Response: During the purification, we did not add Mn²⁺ or other any metal ions. We obtained FKRP crystals with Mg²⁺ or Ba²⁺ ions, which were added to the crystallization buffer (these conditions are described in the Methods section). Unfortunately, we could not obtain FKRP crystals in the presence of MnCl₂. Mn²⁺ is the closest analogue of Mg²⁺ in biological systems, and the two ions have very similar coordination spheres as described in the following references:

Zheng, Heping et al. “CheckMyMetal: a macromolecular metal-binding validation tool.” (2017) *Acta crystallographica. Section D, Structural biology* 73: 223-233. doi:10.1107/S2059798317001061

Bock, C., Katz, A., Markham, G. & Glusker, J. “Manganese as a Replacement for Magnesium and Zinc: Functional Comparison of the Divalent Ions” (1999) *J. Am. Chem. Soc.* 121, 7360–7372. doi:10.1021/ja9906960

Therefore, the coordination spheres of Mg^{2+} and Mn^{2+} are expected to be essentially the same.

In the legend to Figure S4, the word “respectively” was added (page 30).

Figure 3: based on similarities with canonical NTases, the Authors state that CDP-Rbo is likely to interact with the metal in site II but no metal was detected there I reckon. Was any Ba^{2+} observed in site II?

Response: Ba^{2+} was not observed at site II in the Ba and CDP-Rbo bound form (Figure 3C, Supplemental Table S3). To assist readers in their understanding of this matter, we prepared a new figure (Supplemental Figure S8A) which superposes Figures 3C and 3D and referred to it on page 9 line 2.

Upon catalysis, the CDP-Rbo donor is transformed into CMP and the ribitol-P transferred onto dystroglycan. The CMP-bound structure with Mg^{2+} has been solved, but not with Mn^{2+} . However, what is happening next? Is CMP released when a new CDP-Rbo is available to enter the catalytic site? Is it implied that site I is not completely occupied by a metal atom? Again, is Mg^{2+} or Mn^{2+} the metal found in native conditions? In general, an overall synoptic cartoon indicating the two domains (stem and catalytic), the relevant areas (site I and site II) within the catalytic domain in the substrate-free state, the subsequent reaction steps involving CDP-Rbo (donor) and RboP-(phospho-)core M3 (acceptor) as well as the bidimensional chemical structures of donor and acceptor, and possibly highlighting some of the points still open (Mg vs Mn ?), could be helpful especially for the general audience and could be proposed within the Discussion.

Response: Thank you for the comment. As pointed out, we have no experimental data for the metal ions of FKRP in cells. Since we have insufficient data to explain the full reaction cycle of FKRP, it is difficult to provide any concrete speculation as to what happens next. However, our structural and biochemical data do successfully reveal the initial part of the catalytic reaction of FKRP. To clarify the points that we have revealed in this study and open questions, we prepared supplemental Figure S10. As a future

project, we would like to perform a kinetic and structural study to reveal the next step of the reaction, including the release of CMP, and the mechanistic function of the two metal sites, *etc.*

Figure 3F: Why was Asp360 not mutated to Ala too? It should be involved in coordination at site I (weaker interactions with Mg²⁺). Indeed, as reported in Fig. S2, in the fly an alanine residue occupies the same topological position.

Response: We prepared the Asp360Ala mutant and analyzed its enzymatic activity. As expected, the mutant showed no enzymatic activity. Figure 3F was therefore replaced with a new figure that included the results for the Asp360Ala mutant. The main text was also revised as follows:

[Results, page 8 lines 16-17]

“In addition, Asp360, which coordinates the Mg²⁺ at binding site I, is indispensable to the enzymatic activity (Figure 3F).”

[Discussion, page 12 lines 15-18]

“In addition, His42 in ANT(2⁺)-Ia, which corresponds to Asp360 in FKRP, is an essential residue for interacting with the nucleotide donor, ATP [35,36]. This coincides well with our finding that Asp360 in FKRP was essential for the enzymatic activity and interacts with CDP-Rbo via a metal ion in site I.”

[Figure 3 legend (page 16)]

“Enzymatic activities of sFKRP (WT, D360A, D362A, D364A, and D416A) . . . ”

“The CDP-Rbo binding evokes conformational changes” paragraph is written in a sketchy fashion. For example, “These changes were related to each other and interactions were found among them” (interactions between changes?). Is the CDP-Rbo unbound form simply the substrate-free form? Please use there and elsewhere “substrate-free” instead of “unbound”. “The C-terminal fragment in subunit A adopts an alpha helix”, please re-phrase.

Response: As the referee pointed out, the original sentence “These changes were related to each other and interactions were found among them” was ambiguous, so we deleted it [page 9, lines 9-10].

In addition, we followed the referee’s suggestion and replaced the word “unbound” with “substrate-free” at [page 9, line 12], and [Figure 3E legend (page 16)].

Finally, we rephrased the following sentence as recommended (page 9, lines 12-14): “In the substrate-free form, residues from Pro481 to Thr492 of subunits A and C adopt an α helical conformation (Figure 3A) and interact with subunits B and D,

respectively.”

Were all the reported conformational changes (Fig. 3F) identified comparing the CDP-Rbo bound form with Ba²⁺ to the substrate-free form with Ba²⁺, or also with Mg²⁺? In case, did any relevant discrepancies emerge between the substrate-free Ba²⁺ and Mg²⁺ forms, aside from the coordination sites?

Response: First, we found that it was impossible to observe conformational changes of the Mg²⁺-bound form upon CDP-Rbo binding because the Mg²⁺-bound form reacts with CDP-Rbo, leaving the RboP moiety; only the CMP moiety was observed in the Mg²⁺-bound form (page 12, lines 18-20). On the other hand, we were able to observe conformational changes of the Ba²⁺-bound substrate-free form (Figure 3B) upon CDP-Rbo binding (Figure 3C). The changes in the Ba²⁺ binding form were nearly the same as those shown in Figure 3E (superposition of the substrate-free form (Mg²⁺) and CDP-Rbo-bound form (Ba²⁺)). This is reasonable because no significant structural differences were observed between the Mg²⁺- and Ba²⁺-bound substrate free forms (new Supplemental Figure S8B).

It was noteworthy that the position of the CMP moiety of the Ba²⁺-CDP-Rbo complex was nearly the same as that of the Mg²⁺-CMP complex. Therefore, the binding site of the CDP-Rbo seems not to be affected by the metal ion in the active site.

Page 10: please amend to “In addition, we performed a mass spectrometry experiment to determine...”.

Response: We changed the indicated phrase (page 10, line 5 from the bottom) from “we performed another experiment” to “we performed a mass spectrometry experiment”.

Comments of Referee 2

The authors need to clarify the functional importance of FKRP tetramer. Based on the structural and biochemical analyses, the “protomeric dimer” of FKRP is clearly important. However, the functional significance of the tetramer is unclear. However, the authors claim in several places that the FKRP “oligomer” is important, which is vague and not accurate. The authors should either state that the dimer is important, but the significance of the tetramer is unclear, or set out to evaluate the importance of the tetramer by making mutations in the tetramer interface. Does FKRP exist as a dimer or higher-order oligomer in vivo?

Response: We agree with the referee’s comment that the dimer is important, but the significance of the tetramer is unclear. We added the following sentence (page 13, lines 16-18):

“We showed the importance of the protomeric dimer using crystal structures and SEC/enzymatic measurements of the mutants, but the significance of the tetramer remains elusive.”

A previous study reported that FKRP forms a dimer or an oligomer *in vivo* (Alhamidi M et al., 2011, PLoS ONE 6:e22968). We cited this study [26] and in fact also mentioned it in the Introduction section of the original manuscript (page 4, line 28).

The authors need to clarify the functional importance of site I and site II metal ions.

1) *Figure 3B: can the authors show the anomalous difference map for Ba, like the one they showed in Figure 1C for Zn? Can the authors comment on why only one Ba atom is found in this structure, and why it is seen at the weak binding site I, but not the stronger site II?*

Response: Thank you for the important comment. We added an anomalous difference Fourier map for Ba²⁺ in Supplemental Figure S7. The difference of the binding sites between Ba²⁺ and Mg²⁺ could be explained by the difference of the ionic radii of the two ions. The approximate ionic radii of Ba²⁺ and Mg²⁺ are estimated as 1.35 and 0.72 Å, respectively (Shannon, RD (1976) Acta Cryst. A32, 751-767). As shown in Supplemental Table S3, the weak binding site (site I) shows long coordination distances even in the Mg²⁺ binding form. Therefore, it may be possible to accommodate a large Ba²⁺ ion. However, the strong site (site II) has a small coordination sphere and it cannot accommodate a Ba²⁺ ion. We added the following sentences (page 7 bottom to page 8 top) to describe the different binding sites between Mg²⁺ and Ba²⁺:

“Since site I has long coordination distances (Table S3), it seems to be capable of accommodating Ba²⁺ with ionic radius of approximately 1.4 Å, which is larger than the

radii size of 0.7 Å of Mg²⁺ (Figures 3B, S7 and S8B). However, site II has significantly shorter coordination distances than site I, and thus it seems to be impossible to bind Ba²⁺ without destroying the coordination sphere.”

2) *Figure 3C and 3D: based on these two structures, the authors suggest that “the metal ion at site II, which could not be observed in the crystal structure, also interacts with the donor molecule CDP-Rbo” (Page 8). This is not very clear in the current presentation. Can the authors overlay the two structures, and describe the distances of site II metal to CDP-Rbo atoms if it is present?*

Response: Thank you for the comment. We prepared a new figure that superposes the two structures to clarify the situation (Supplemental Figure S8A), and we referred to this figure on page 9 line 2. The distances between Mg²⁺ and CDP-Rbo are described in the legend of Supplemental Figure S8A (page 31).

3) *Figure 3F: Asp362 and Asp364 are involved in binding to both site I and site II metal ions, so their mutations could affect metal binding to both sites. Have the authors made the D360A mutant and tested its activity?*

Response: Thank you for the comment. We prepared the Asp360Ala mutant and analyzed its enzymatic activity. As expected, the mutant showed no enzymatic activity. Figure 3F was therefore replaced with a new figure that included the results for the Asp360Ala mutant. The main text was also revised as follows:

[Results, page 8 lines 16-17]

“In addition, Asp360, which coordinates the Mg²⁺ at binding site I, is indispensable to the enzymatic activity (Figure 3F).”

[Discussion, page 12 lines 15-18]

“In addition, His42 in ANT(2’)-Ia, which corresponds to Asp360 in FKRP, is an essential residue for interacting with the nucleotide donor, ATP [35,36]. This coincides well with our finding that Asp360 in FKRP was essential for the enzymatic activity and interacts with CDP-Rbo via a metal ion in site I.”

[Figure 3 legend (page 16)]

“Enzymatic activities of sFKRP (WT, D360A, D362A, D364A, and D416A) . . . ”

4) *In the Discussion section, the authors suggest that FKRP has a similar catalytic mechanism as NTase, with the site II metal playing a catalytic function. This section needs to be consolidated, and some figures are necessary to illustrate the active site comparison between FKRP and NTase.*

Response: We made a new figure (Supplemental Figure S1B) which compares the active sites between FKRP and NTase. We rewrote the first paragraph of the Discussion section as follows (pages 11-12) to refer to the figures (Supplemental Figure S1A,B) and indicated some specific residues:

“The structure of the core region of the catalytic domain is similar to that in NTase family proteins. Most known members of the NTase superfamily transfer NMP from NTP to an OH group of the acceptor molecules and release pyrophosphate (PPi) (NTP → NMP + PPi) [29]. In contrast, FKRP releases CMP from CDP-Rbo and transfers a RboP moiety to an OH group of the acceptor, RboP-(phospho-)core M3 [11]. Both reactions are dehydration/condensation reactions with hydroxyl and phosphate groups, but the leaving group of the donor substrate in FKRP is different from that in typical NTase, even if FKRP has common sequence motifs, [DE]h[DE]h and h[DE]h (where h indicates a hydrophobic amino acid), and a protein fold of NTase [29] (Figures S1A and S2). Our structural analysis revealed a similarity between the catalytic reactions of FKRP and NTase. NTase requires two divalent metal ions, such as Mn^{2+} and Mg^{2+} , which interact with the donor and/or acceptor molecule in the active site (Figure S1B, right panel) [36]. In the crystal structure of ANT(2’)-Ia, a representative NTase, one metal ion at binding site I coordinates two aspartic acid residues (Asp44 and Asp46) and the diphosphate moiety of the donor molecule. The other metal ion at binding site II coordinates three aspartic acid residues (Asp44, Asp46 and Asp86) and the OH group of the acceptor molecule. Based on the results for ANT(2’)-Ia and the DNA polymerase β subunit [36,37], it has been proposed that the third aspartic acid residue (Asp86) interacting with a metal ion at site II acts as an acidic base and leads to electrophilic polarization of the OH group by coordinating the metal ion. The crystal structure of the sFKRP-CMP complex revealed a similarity of the metal ion coordination sites to those of NTase; one Mg^{2+} at site I coordinates Asp362 and Asp364, while the other Mg^{2+} at site II coordinates Asp362, Asp364, and Asp416 (Figures 3D and S1B). This similarity suggests that FKRP and NTase share a catalytic mechanism.”

We inserted the following sentence at the start of the second paragraph of the Discussion section (page 12, lines 6-7).

“Our crystal structure suggests that Asp416 would interact with the Rbo moiety of an acceptor glycopeptide via the metal ion at site II.”

We inserted the following sentences into the middle of the second paragraph of the Discussion section (page 12, lines 15-18):

“In addition, His42 in ANT(2’)-Ia, which corresponds to Asp360 in FKRP, is an essential residue for interacting with the nucleotide donor, ATP [35,36]. This fact

coincides well with our results that Asp360 in FKRP was essential for the enzymatic activity and interacts with CDP-Rbo via a metal ion in site I.”

The structure with the RboP-(phosphor-)coare M3 peptide bound is the most important result in this paper. It is unfortunate that the electron density for several parts of the acceptor molecule are unclear, especially for the Rbo moiety. How do the authors know that an intact acceptor glycopeptide, rather than a degradation product of this molecule, is present in the crystal? Have the authors tried to use a catalytically dead mutant of FKRP to obtain the complex structure?

Response: Thank you for the comment. When the enzymatic reaction of FKRP was analyzed in the presence of Mg^{2+} , Mn^{2+} or Ba^{2+} using mass spectrometry, we could not detect a molecular species that lacks the Rbo moiety from the acceptor. FKRP does not hydrolyze the Rbo residue from the acceptor glycopeptide, suggesting that the acceptor is intact and the Rbo moiety of the acceptor is mobile in the crystal. In addition, we determined this crystal in the presence of Ba^{2+} . As we described in the main text and Supplemental Figure S4, the Ba^{2+} binding form of FKRP has no enzymatic activity. Therefore, the experiments were carried out using an inactive enzyme that is equivalent to an enzymatically dead mutant.

The presentation of Figure 4 is also confusing and needs to be thoroughly edited.

Response: We revised Figure 4 for clarity.

1) Figure 4A: the authors should label the CDP-Rbo and the phosphor-core M3. Also, the CDP-rbo appears all purple here, not “yellow and purple” as described in the legend, which is confusing.

Response: Following the referee’s suggestion, we added labels for the molecules in Figure 4A. There are two CDP-Rbo molecules in Figure 4A; they are shown by yellow and purple colors, respectively.

2) Figure 4B: the authors should consider showing the Fo-Fc map for both the CDP-Rbo and the phosphor-core M3 molecules, since this is a new (and the most important) structure.

Response: Following the referee’s suggestion, we prepared a new figure with Fo-Fc maps for the right half of Figure 4B.

3) *Figure 4C: The interaction between H252/K256 and phospho-6Man should be indicated with dashed lines. Also, are the labels for GalNAc and GlcNAc swapped? “RboP-3GalNAc β 1-3GlcNAc β 1-4(phospho-6)Man α 1” is described in the text (Page 9), so GlcNAc should be in the middle. Also related to this, “the N-acetyl group of the GlcNAc moiety interacts with Thr299 and His412” (Page 9)—should be the N-acetyl group of GalNAc instead? This sentence is also not in the correct place—both the sentence before it and after it are describing the phosphate group of RboP.*

Response: Following the referee’s suggestion, we added dashed lines to show the interactions between the phospho-core M3 moiety and the protein. As the referee pointed out, the labels for GalNAc and GlcNAc in the Figure were swapped. We corrected panel C in Figure 4 and the corresponding passage in the Results (page 10, line 11). Thank you for the comments.

Based on the structure of FKR, can the authors make a structural model for Fukutin, and comment on their similarities and differences, especially in the regions that bind to the acceptor molecules? This would shed light on their different substrate specificity.

Response: We prepared a model structure of Fukutin (FKTN) based on the FKR structure using a SWISS-model (Supplemental Figure S9), and added a paragraph to the Discussion section to describe the structure of the model (page 12 bottom to page 13 top):

“We compared FKR and FKTN, which show different substrate specificities, by their primary sequences. The amino acid sequence identity between their stem domains (45 to 287 in FKR and 44 to 248 in FKTN) was very low (8.2%), suggesting that the molecular mechanism, particularly the recognition mechanism of the acceptor molecule, would be different. On the other hand, the sequence identity at the catalytic domains was 20% (288 to 451 in FKR and 249 to 461 in FKTN). Thus, the structure of the catalytic domain of FKTN would be similar with that of FKR; the structural similarity of their catalytic domains was also suggested by comprehensive sequence analysis [29]. We therefore made a homology model of the catalytic domain of FKTN based on the FKR structure by using a Swiss-model (Figure S9) [38]. The model showed that FKTN has no zinc finger loop and Arg residue corresponding to Arg295 in FKR, while the three Asp residues corresponding to Asp362, Asp364 and Asp416 in FKR are conserved. These facts suggest that the sugar acceptor recognition (i.e., core M3 recognition) is completely different between FKR and FKTN, although the metal and sugar donor recognitions were similar to each other.”

38. Waterhouse A et al. (2018) *Nucleic Acids Res* 46: W296-W303.

Additional Corrections

1) Structural data were checked again according to the suggestions of the PDB annotators. We then revised supplemental tables S1 and S2. The buried surface area of the protomeric dimer was re-calculated, from 1,652 Å² to 1,642 Å² (page 5, bottom of the first paragraph). Refinement statistics from Ba-SAD data were added to supplemental table S1. The refined data were used for making anomalous difference Fourier map around Barium ion in supplemental Figure S7.

2) page 7, bottom

A typo was corrected:

“at site II of subunit B”

3) page 8, line 3 from the bottom

Duplicated words were deleted:

“the α phosphate residue of CDP-Rbo seems to interact with the Ba²⁺ ion in site I ~~and the RboP moiety~~ (Figure 3C, left grey loop).”

4) page 10, line 5

A typo was corrected:

“phospho-3GalNAc β 1-3GlcNAc β 1-4(phospho-6)Man”

5) Figure 4, legend (page 16)

A typo was corrected (three places):

“phospho-(phospho-)core M3”

6) Acknowledgements (page 28, line 3 from the bottom)

A missed grant number was added:

“JP19H05648 (to T.E., K.K. and R.K.)”

7) The references were renumbered to accommodate the new additions (see the list below). The changed reference numbers are highlighted yellow in the revised manuscript.

Old Ref. No	New Ref. No	First author, Year, Title
-	18	de Paula (2003) Asymptomatic carriers for homozygous novel mutations in the FKRP gene: the other end of the spectrum.

-	19	Beltran-Valero de Bernabé D (2004) Mutations in the FKRP gene can cause muscle-eye-brain disease and Walker-Warburg syndrome.
-	20	Brockington M (2001) Mutations in the fukutin-related protein gene (FKRP) cause a form of congenital muscular dystrophy with secondary laminin alpha2 deficiency and abnormal glycosylation of alpha-dystroglycan.
-	21	Esapa CT (2002) Functional requirements for fukutin-related protein in the Golgi apparatus.
-	22	Esapa CT (2005) Fukutin-related protein mutations that cause congenital muscular dystrophy result in ER-retention of the mutant protein in cultured cells.
-	23	Henriques SF (2019) Functional and cellular localization diversity associated with Fukutin-related protein patient genetic variants.
18	24	
-	25	Gerin I (2016) ISPD produces CDP-ribitol used by FKTN and FKRP to transfer ribitol phosphate onto alpha-dystroglycan.
19	26	
20	27	
21	28	
22	29	
23	30	
24	31	
25	32	
26	33	
-	34	Mercuri E (2003) Phenotypic spectrum associated with mutations in the fukutin-related protein gene.
27	35	
28	36	
29	37	
-	38	Waterhouse A (2018) SWISS-MODEL: homology modelling of protein structures and complexes.
30	39	
31	40	
32	41	
33	42	
34	43	
35	44	
36	45	
37	46	
38	47	
39	48	
40	49	
41	50	
42	51	
43	52	
44	53	

REVIEWERS' COMMENTS:

Reviewer #1 (Remarks to the Author):

I reckon that my comments were addressed properly and the manuscript has improved.

With hindsight, it might be really interesting the observation that L276I is still able to form higher MW aggregates (Fig.2C), as perhaps this "aggregation ability" (potentially leading to octamers?) could help preserving its enzymatic activity (that is reduced only by 50% with respect to WT). As stated by the Authors in their reply, the patients with Leu276Ile mutation developed the disease in the second decade of life or later (milder LGMD2I).

Just a technicality and a suggestion. Given that S221R and L276I were not secreted into the culture media (as stated in the "Enzymatic assay for sFKRP" paragraph), were cell lysates also used for the SEC experiment reported in Fig. 2C? This could be specified in the "Oligomerization analysis of sFKRP mutants" methods paragraph. Can the Authors report (in the fig legend and methods) which kind of gel analysis (followed by Western Blotting) was used in Fig. 2C (a native gel I presume)? Perhaps, in Methods, although obvious they could also add that the gels used in immunoblots to quantitate the proteins (Fig.2D,E, Fig. 3F and Fig. 4D) were SDS-PAGE.

I do not need to see again the manuscript and I have no additional comments.

Andrea Brancaccio

Reviewer #2 (Remarks to the Author):

The authors have responded to most of my comments. It is unfortunate that the Rbo moiety in the complex structure can not be observed. Nevertheless, the revised manuscript is significantly improved and suitable for publication in my opinion.

Responses to the Reviewer#1's Comments/Suggestions:

Comment:

With hindsight, it might be really interesting the observation that L276I is still able to form higher MW aggregates (Fig.2C), as perhaps this “aggregation ability” (potentially leading to octamers?) could help preserving its enzymatic activity (that is reduced only by 50% with respect to WT). As stated by the Authors in their reply, the patients with Leu276Ile mutation developed the disease in the second decade of life or later (milder LGMD2I).

Response:

Thank you for the suggestion. We agree the possibility, and added the following sentence (page 8, lines 15-16):

“In the case of Leu276Ile mutant, higher molecular weight aggregates may help to partially preserve the enzymatic activity.”

Comment:

Just a technicality and a suggestion. Given that S221R and L276I were not secreted into the culture media (as stated in the “Enzymatic assay for sFKRP” paragraph), were cell lysates also used for the SEC experiment reported in Fig. 2C? This could be specified in the “Oligomerization analysis of sFKRP mutants” methods paragraph. Can the Authors report (in the fig legend and methods) which kind of gel analysis (followed by Western Blotting) was used in Fig. 2C (a native gel I presume)? Perhaps, in Methods, although obvious they could also add that the gels used in immunoblots to quantitate the proteins (Fig.2D,E, Fig. 3F and Fig. 4D) were SDS-PAGE.

Response:

Thank you for the comment.

Since S221R and L276I were scarcely secreted into the culture media, we used them for the SEC experiment (Fig. 2C). However, since the amounts were not enough to measure the enzymatic activities, we used the cell lysates for the activity measurement. We corrected the “Enzymatic assay for sFKRP” section in the Methods as follows (page 19, end of the page):

“Alternatively, because the Ser221Arg and Leu276Ile mutants were scarcely secreted to the culture media, ”

We carried out SDS-PAGE in all electrophoresis experiments in the manuscript (Figs 2C,D,E, 3F, and 4D). We added the information to the legend of Figure 2C, and the “Enzymatic assay for sFKRP” section of the Methods (for Figs. 2D,E, 3F, and 4D).